# GENERATIVE PRETRAINING FOR BLACK-BOX OPTIMIZATION

## ABSTRACT

Many problems in science and engineering involve optimizing an expensive black-box function over a high-dimensional space. For such black-box optimization (BBO) problems, we typically assume a small budget for online function evaluations, but also often have access to a fixed, offline dataset for pretraining. Prior approaches seek to utilize the offline data to approximate the function or its inverse but are not sufficiently accurate far from the data distribution. We propose BONET, a generative framework for pretraining a novel black-box optimizer using offline datasets. In BONET, we train an autoregressive model on fixed-length trajectories derived from an offline dataset. We design a sampling strategy to synthesize trajectories from offline data using a simple heuristic of rolling out monotonic transitions from low-fidelity to high-fidelity samples. Empirically, we instantiate BONET using a causally masked Transformer (Radford et al., 2019) and evaluate it on Design-Bench (Trabucco et al., 2022), where we rank the best on average, outperforming state-of-the-art baselines.

## 1   INTRODUCTION

Many fundamental problems in science and engineering, ranging from the discovery of drugs and materials to the design and manufacturing of hardware technology, require optimizing an expensive black-box function in a large search space (Larson et al., 2019; Shahriari et al., 2016). The key challenge here is that evaluating and optimizing such a black-box function is typically expensive, as it often requires real-world experimentation and exploration of a high-dimensional search space.

Fortunately, for many such black-box optimization (BBO) problems, we often have access to an offline dataset of function evaluations. Such an offline dataset can greatly reduce the budget for online function evaluation. This introduces us to the setting of offline BBO. A key difference exists between the offline BBO setting and its online counterpart; in offline BBO, we are **not** allowed to actively query the black-box function during optimization, unlike in online BBO where most approaches (Snoek et al., 2012; Shahriari et al., 2016) utilize iterative online solving. One natural approach for offline BBO would be to train a surrogate (forward) model that approximates the black-box function using the offline data. Once learned, we can perform gradient ascent on the input space to find the optimal point. Unfortunately, this method does not perform well in practice because the forward model can incorrectly give sub-optimal and out-of-domain points a high score (see Figure 1a). To mitigate this issue, COMs (Trabucco et al., 2021) learns a forward mapping that penalizes high scores on points outside the dataset, but this can have the opposite effect of not being able to explore high fidelity points that are far from the dataset. Further, another class of recent approaches (Kumar & Levine, 2020; Brookes et al., 2019; Fannjiang & Listgarten, 2020) propose a conditional generative approach that learns an inverse mapping function values to the points. For effective generalization, such a mapping needs to be highly multimodal for high-dimensional functions, which in itself presents a challenge for current approaches.

We propose **B**lack-box **O**ptimization **Net**works (BONET), a new generative framework for pretraining black-box optimizers on offline datasets. Instead of approximating the surrogate function (or its inverse), we seek to approximate the dynamics of online black-box optimizers using an autoregressive sequence model. Naively, this would require access to several trajectory runs of different black-box optimizers, which is expensive or even impossible in many cases. Our key observation is that we can synthesize synthetic trajectories comprised of offline points that mimic empirical characteristics

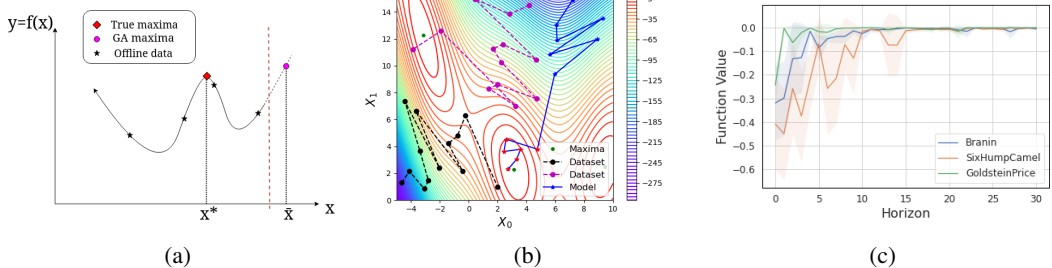

Figure 1: (a) Example of offline BBO on toy 1D problem. Here, the domain ends at the red dashed line. Thus, the correct optimal value is $\mathbf{x}^*$, whereas gradient ascent on the fitted function will output out of the domain point $\bar{\mathbf{x}}$. (b) Example trajectory on the 2D-Branin function. The dotted lines denote the trajectories in our offline dataset, and the solid line refers to our model trajectory, with low-quality blue points and high-quality red points. (c) Function values of trajectories generated by a simple gaussian process (GP) based BayesOpt model on several synthetic functions.

of online BBO algorithms, such as BayesOpt. While one could design many characteristic properties, we build off an empirical observation related to the function values of the proposed points. In particular, averaged over multiple runs, online black-box optimizers (e.g., BayesOpt) tend to show improvements in the function values of the proposed points (Bijl et al., 2016), as shown in Figure 1c. While not exact, we build on this observation to develop a sorting heuristic that constructs synthetic trajectories consisting of offline points ordered monotonically based on their ascending function values. Even though such a heuristic does not apply uniformly for the trajectory runs of all combinations of black-box optimizers and functions, we show that it is simple, scalable, and quite effective in practice.

Further, we augment every offline point in our trajectories with a *regret budget*, defined as the cumulative regret of the trajectory starting at the current point until the end of the trajectory. We train BONET to generate trajectories conditioned on the regret budget of the first point of the trajectory. Thus, at test time, we can generate good candidate points by rolling out a trajectory with a low regret budget. Figure 1b shows an illustration.

We evaluate our method on several real-world tasks in the Design-Bench (Trabucco et al., 2022) dataset. These tasks are based on real-world problems such as robot morphology optimization, DNA sequence optimization, and optimizing superconducting temperature of materials, all of which requires searching over a high-dimensional search space. We achieve a normalized mean score of **0.772** and an average rank of **2.4** across all tasks, outperforming the next best baseline, which achieves a rank of 3.7.

## 2 PRETRAINING BLACK-BOX OPTIMIZERS VIA BONET

### 2.1 PROBLEM STATEMENT

Let $f : \mathcal{X} \to \mathbb{R}$ be a black-box function, where $\mathcal{X} \subseteq \mathbb{R}^d$ is an arbitrary $d$-dimensional domain. In black-box optimization (BBO), we are interested in finding the point $\mathbf{x}^*$ that maximizes $f$:

$$\mathbf{x}^* \in \underset{\mathbf{x} \in \mathcal{X}}{\arg\max} \, f(\mathbf{x}) \tag{1}$$

Typically, $f$ is expensive to evaluate and we do not assume direct access to it during training. Instead, we have access to an offline dataset of $N$ previous function evaluations $\mathcal{D} = \{(\mathbf{x}_1, y_1), \cdots, (\mathbf{x}_N, y_N)\}$, where $y_i = f(\mathbf{x}_i)$. For evaluating a black-box optimizer post-training, we allow it to query the black-box function $f$ for a small budget of $Q$ queries and output the point with the best function value obtained. This protocol follows prior works in offline BBO (Trabucco et al., 2021; 2022; Kumar & Levine, 2020; Brookes et al., 2019; Fannjiang & Listgarten, 2020).

**Overview of BONET** We illustrate our proposed framework for offline BBO in Figure 2 and Algorithm 1 . BONET consists of 3 sequential phases: trajectory construction, autoregressive modelling,

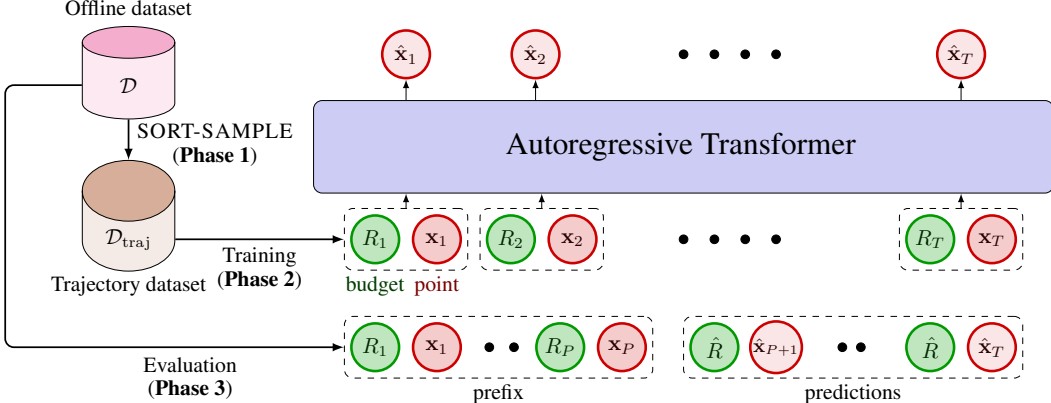

Figure 2: Schematic for BONET. In Phase 1, we construct a trajectory dataset $\mathcal{D}_{\text{traj}}$ using SORT-SAMPLE. In Phase 2, we learn an autoregressive model for $\mathcal{D}_{\text{traj}}$. In Phase 3, we condition the model on an offline prefix sequence and unroll it further to obtain candidate proposals $\hat{\mathbf{x}}_{P+1:T}$.

roll-out evaluation. In Phase 1 (Section 2.2), we transform the offline dataset $\mathcal{D}$ into a trajectory dataset $\mathcal{D}_{\text{traj}}$. This is followed by Phase 2 (Section 2.3), where we train an autoregressive model on $\mathcal{D}_{\text{traj}}$. Finally, we evaluate the model by rolling out $Q$ candidate points in Phase 3 (Section 2.4).

## 2.2 PHASE 1: CONSTRUCTING TRAJECTORIES

Our key motivation in BONET is to train a model to mimic the sequential behavior of online black-box optimizers. However, the difficulty is that we do not have the ability to generate trajectories by actively querying the black-box function during training. In BONET, we overcome this difficulty by synthesizing trajectories purely from an offline dataset based on two guiding desiderata.

---

**Algorithm 1 B**lack-box **O**ptimization **Net**works (BONET)

    **Input** Offline dataset $\mathcal{D}$, Evaluation Regret Budget $\hat{R}$, Prefix length P, Query budget $Q$, Trajectory length $T$, num_trajs, Smoothing parameter $K$, Temperature $\tau$, Number of bins $N_B$
    **Output** A set of proposed candidate points $\mathbf{X}$ with the constraint $|\mathbf{X}| \leq Q$
 1: ▷ **Phase** 1: SORT-SAMPLE
 2: Construct bins $\{B_1, \cdots, B_{N_B}\}$ from $\mathcal{D}$, each bin covering equal $y$-range, as described in 2.2
 3: Calculate the scores $(n_1, n_2, \cdots, n_{N_B})$ for each bin using $K$ and $\tau$
 4: $\mathcal{D}_{\text{traj}} \leftarrow \phi$
 5: **for** $i = 1, \cdots,$ num_trajs **do**
 6:     Uniformly randomly sample $n_i$ points from $B_i$ and concatenate them to construct $\mathcal{T}$
 7:     Sort $\mathcal{T}$ in the ascending order of the function value
 8:     Represent $\mathcal{T}$ as $(R_1, \mathbf{x}_1, R_2, \mathbf{x}_2, \cdots, R_T, \mathbf{x}_T)$, following equation 3, and append $\mathcal{T}$ to $\mathcal{D}_{\text{traj}}$
 9: **end for**
10: ▷ **Phase** 2: Training
11: Train the model $g_\theta$ to maximize the log-likelihood of $\mathcal{D}_{\text{traj}}$ using the loss in equation 4
12: ▷ **Phase** 3: Evaluation
13: Construct a trajectory $\mathcal{T}'$ from $\mathcal{D}$ following Phase 1, and delete the last $T - P$ points
14: Calculate $R_t$ and feed $(R_t, \mathbf{x}_t)$ to $g_\theta$ sequentially, $\forall t = 1, \cdots, P$
15: Roll-out $g_\theta$ autoregressively while feeding $R_t = \hat{R}, \forall t = P + 1, \cdots, T$
16: $\mathbf{X}$ is the set of last $\min(Q, T - P)$ rolled-out points.

---

First, the procedure for synthesizing trajectories should efficiently scale to high-dimensional data points and large offline datasets. Second, each trajectory should mimic characteristic behaviors commonly seen in online black-box optimizers. We identify one such characteristic of interest. In particular, we note that the moving average of function values of points proposed by such black-box optimizers tends to improve over the course of their runs barring local perturbations (e.g., due to

exploration). While exceptions can and do exist, this phenomena is commonly observed in practice for optimizers such as BayesOpt (Bijl et al., 2016). We also illustrate this behavior in Figure 1c for some commonly used test functions optimized via BayesOpt.

**Sorted Trajectories** We propose to satisfy the above desiderata in BONET by following a sorting heuristic. Specifically, given a set of $T$ offline points, we construct a trajectory of length $T$ by simply sorting the points in ascending order from low to high function values. We note that sorting is just a heuristic we use for constructing synthetic trajectories from the offline dataset, and this behavior may not be followed by any general optimizer over any arbitrary functions. We also perform ablations on different heuristics in Appendix C.1. Further, we note that sorting does not provide any guidance on the rate or the relative spacing between the points i.e., how fast the function values increase. This rate is important for controlling the sample budget for black-box optimization. Next, we discuss a sampling strategy for explicitly controlling this rate.

**Sampling Strategies for Offline Points** So far, we have proposed a simple heuristic for transitioning a set of offline points into a sorted trajectory. To obtain these offline trajectory points from the offline dataset, one default strategy is to sample uniformly at random $T$ points from $\mathcal{D}$ and sort them. However, we found this strategy to not work well in practice. Intuitively, we might expect a large volume of the search space to consist of points with low-function values.

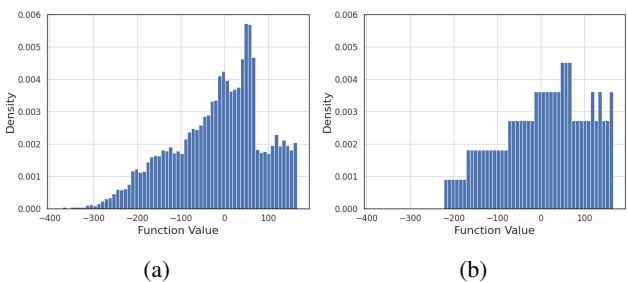

(a)                                    (b)

Figure 3: Plots showing the distribution of function values in the offline dataset $\mathcal{D}$ (left) and the trajectories in $\mathcal{D}_{\text{traj}}$ (right) for the Ant Morphology benchmark (Trabucco et al., 2022). Notice how the overall density of points with high function values is up-weighted post our re-weighting.

Thus, if our offline dataset is uniformly distributed across the domain, the probability of getting a high quality point will be very low with a uniform sampling strategy. To counter this challenge, we propose a 2 step sampling strategy based on binning followed by importance reweighting. Our formulation is motivated by a similar strategy proposed by Kumar & Levine (2020) for loss reweighting. First, we use the function values to partition the offline dataset $\mathcal{D}$ into $N_B$ bins of equal-width, i.e., each bin covers a range of equal length. Next, for each bin, we assign a different sampling probability, such that (a) bins where the average function value is high are more likely to be sampled, and (b) bins with more points are sampled more often. The former helps minimize the budget, whereas the latter ensures diversity in trajectories. Based on these two criteria, the score $s_i$ for a bin $B_i$ is given as:

$$s_i = \frac{|B_i|}{|B_i| + K} \exp\left(\frac{-|\hat{y} - y_{b_i}|}{\tau}\right) \tag{2}$$

where $\hat{y}$ is the best function value in the dataset $\mathcal{D}$, $|B_i|$ refers to the number of points in the $i^{th}$ bin, and $y_{b_i}$ is the midpoint of the interval corresponding to the bin $B_i$. Here, the first term $\frac{|B_i|}{|B_i|+K}$ allows us to assign a higher weight to the larger bins with smoothing. The second term gives higher weight to the good bins using an exponential weighting scheme. More details about $K$ and $\tau$ can be found in Appendix B. Finally, we use these scores $s_i$ to proportionally sample $n_i$ points from bin $B_i$ where $n_i = T\left\lfloor \frac{s_i}{\sum_j s_j} \right\rfloor$ for $i \in \{2, \cdots, N_B\}$ and $n_1 = T - \sum_{i>1} n_i$, making the overall length of the trajectories equal to $T$. In Figure 3, we illustrate the shift in distribution of function values due to our sampling strategy. We refer to the combined strategy of sampling and then sorting as SORT-SAMPLE in Figure 2 and Algorithm 1.

**Augmenting Trajectories With Regret Budgets** Our sorted trajectories heuristically reflect rollouts of implicit black-box optimizers. However they do not provide us with information on the rate at which a trajectory approaches the optimal value. A natural choice for such a knob would be the cumulative regret. Moreover, as we shall show later, cumulative regret provides BONET a simple and effective knob to generalize outside the offline dataset during the evaluation phase.

Hence, we propose to augment each data point $\mathbf{x}_i$ in our trajectory with a Regret Budget (RB). The RB $R_i$ at timestep $i$ defined as the cumulative regret of the trajectory, starting at point $\mathbf{x}_i$: $R_i = \sum_{j=i}^{T}(f(\mathbf{x}^*) - f(\mathbf{x}_j))$. Intuitively, a high (low) value for $R_i$ is likely to result in a high (low) budget for the model to explore diverse points. Note, we are only assuming knowledge of an estimate for $f(\mathbf{x}^*)$ (and not $\mathbf{x}^*$). Thus, each trajectory in our desired set $\mathcal{D}_{\text{traj}}$ can be represented as:

$$\mathcal{T} = (R_1, \mathbf{x}_1, R_2, \mathbf{x}_2, \cdots, R_T, \mathbf{x}_T) \tag{3}$$

We will refer to $R_1$ as Initial Regret Budget (IRB) henceforth. This will be of significance for evaluating our model in Phase 3 (Section 2.4), as we can specify a low IRB to induce the model to output points close to the optima.

## 2.3 PHASE 2: TRAINING AN AUTOREGRESSIVE GENERATIVE MODEL

Given our trajectory dataset, we design our BBO agent as a conditional autoregressive model and train it to maximize the likelihood of trajectories in $\mathcal{D}_{\text{traj}}$. More formally, we denote our model parameterized by $\theta$ as $g_\theta(\mathbf{x}_t | \mathbf{x}_{<t}, R_{\leq t})$, where by $k_{<t}$ we mean the set $\{k_1, \cdots, k_{t-1}\}$. Here, $\mathbf{x}_i$ are the sequence of points in a trajectory, and $R_i$ refers to the regret budget at timestep $i$.

Building on recent advances in sequence modeling (Vaswani et al., 2017; Brown et al., 2020; Radford et al., 2019) , we instantiate our model with a causally masked transformer and train it to maximize the likelihood of our trajectory dataset $\mathcal{D}_{\text{traj}}$.

$$\mathcal{L}(\theta; \mathcal{D}_{\text{traj}}) = \mathop{\mathbb{E}}_{\mathcal{T} \sim \mathcal{D}_{\text{traj}}} \left[ \sum_{i=1}^{T} \log g_\theta(\mathbf{x}_i | \mathbf{x}_{<i}, R_{\leq i}) \right] \tag{4}$$

In practice, we translate this loss to the mean squared error loss for a continuous $\mathcal{X}$ (equivalent to a Gaussian $g_\theta$ with fixed variance), and cross entropy loss for a discrete $\mathcal{X}$.

## 2.4 PHASE 3: EVALUATION ROLLOUT OF FINAL CANDIDATES

Once trained, we can use our BBO agent to directly output new points as its candidate guesses for maximizing the black-box function. We do so by rolling out evaluation trajectories from our model. Each trajectory will be subdivided into a *prefix subsequence* and a *prediction subsequence*. The prefix subsequence consists of $P < T$ points sampled from our offline dataset as before. These prefix points provide initial warm-up queries to the model. Thereafter, we rollout the prediction subsequence consisting of $T - P$ points by sampling from our autoregressive generative model.

**Setting Regret Budgets** One key question relates to setting the regret budget at the start of the suffix subsequence. It is not preferable to set it to $R_{P+1}$ of the sampled trajectory, as doing so will lead to a slow rate of reaching high-quality regions similar to the one observed in the training trajectories. This will not allow the suffix to generalize beyond the offline dataset.

Alternatively, we initialize it to a low value in BONET to accelerate the trajectory towards good points following a prefix subsequence. We refer to this low value as Evaluation RB and denote it as $\hat{R}$ in Figure 2 and in Section 3. Thereafter, we keep the RB for the suffix subsequence fixed ($\hat{R}$), as the agent is expected to be already in a good region. Moreover, updating the RB here would require sequential querying to the function $f$, which can be prohibitive. Thus, our evaluation protocol can generate a set of candidate queries purely in an offline manner. In practice, we also find it helpful to split the $Q$ candidates among a few (and not 1) small $\hat{R}$ values, each with a different prefix.

## 3 EXPERIMENTAL EVALUATION

We first empirically evaluate BONET for optimizing a synthetic 2D function, Branin, in order to analyze its working and probe the various components. Next, we perform large-scale benchmarking and experiments on Design-Bench (Trabucco et al., 2022), a suite of offline BBO tasks based on real-world problems.

## 3.1 BRANIN TASK

Branin is a well-known benchmark function for evaluating optimization methods. It is a 2D function evaluated on the ranges $x_1 \in [-5, 10]$ and $x_2 \in [0, 15]$:

$$f_{br}(x_1, x_2) = a(x_2 - bx_1^2 + cx_1 - r)^2 + s(1 - t)\cos x_1 + s \qquad (5)$$

where $a = -1$, $b = \frac{5.1}{4\pi^2}$, $c = \frac{5}{\pi}$, $r = 6$, $s = -10$, and $t = \frac{1}{5\pi}$. In this square region, $f_{br}$ has three global maximas, $(-\pi, 12.275)$, $(\pi, 2.275)$, and $(9.42478, 2.475)$; with the maximum value of $-0.397887$. Figure 1b shows an illustration of the function contours. For offline optimization we uniformly sample $N = 5000$ points in the domain, and remove the top 10%-ile (according to the function value) from this set to remove points close to the optima to make the task more challenging. We then construct 400 trajectories of length 64 each according to the SORT-SAMPLE strategy.

During the evaluation, we initialize four trajectories with a prefix length of 32 and unroll them for an additional 32 steps, and output the best result, thus consuming a query budget of 128. As we see in Table 1, BONET successfully generalizes beyond the best point in our offline dataset. We also report numbers for a gradient ascent baseline, which uses the offline dataset to train a forward model (a 2 layer NN) mapping $\mathbf{x}$ to $y$ and then performs gradient ascent on $\mathbf{x}$ to infer its optima. Next, we perform ablations to understand the effect of the Evaluation RB $\hat{R}$ and prefix length $P$ on our rolled-out trajectories.

Table 1: Best function value achieved by each method on Branin task. We report mean and standard deviation averaged over 5 runs. Gradient ascent performs poorly because on many initialization points, the trajectories escape out of the square domain.

| OPTIMA | $\mathcal{D}$ (best) | BONET | Grad. Ascent |
|---|---|---|---|
| $-0.398$ | $-6.119$ | $\mathbf{-1.79 \pm 0.843}$ | $-3.953 \pm 4.258$ |

**Impact of $\hat{\mathbf{R}}$** Figures 4a and 4b shows rolled-out trajectories for our model for different $\hat{R}$ values, with prefix lengths 16 and 32. We see that low $\hat{R}$ rolls out higher quality points compared to high $\hat{R}$. To verify our semantics of regret budget as a knob for controlling the rate at which the model accelerates to high-quality points, in the Figure 4c, we also plot trajectories where we update the RB values in the suffix. We stop the roll-out if RB becomes non-positive. It is evident that for smaller $\hat{R}$, the agent quickly accelerates to high-quality regions, whereas for high $\hat{R}$, it gradually shifts to high-quality points. This shows how $\hat{R}$ controls the rate of transition from low to high-quality points.

To further check whether our model has learned to generate a sequence having cumulative regret close to the initial RB $R_1$, we plot $R_1$ vs the cumulative regret of a full rolled-out sequence in Figure 5a. We observe that the curve is close to the desired ideal line $y = x$. Notice that the range of $R_1$ values of $\mathcal{D}_{\text{traj}}$ is quite narrow, but BONET is able to generalize well to a much wider range, allowing it to propose points even better than the dataset. During training, the model has only seen low RB values towards the end of the trajectories. However, the powerful *stitching* ability (Chen et al., 2021) of the model allows it to roll out a novel trajectory having low cumulative regret when conditioned on low unseen $R_1$ values. Finally, since in BBO our goal is to find the best point, we also plot the best rolled-out point across a trajectory versus $\hat{R}$ in Figure 5b while keeping the prefix sequence fixed. As expected, we observe a decreasing trend, justifying our choice of small $\hat{R}$ values.

**Impact of Prefix Length** Figure 5c shows the obtained best function values for different prefix lengths, averaged over multiple $\hat{R}$ values, with same query budget $Q = 32$. As expected, we see an increasing trend in the best function value. We also observe a decreasing variance, indicating that the trajectory roll-outs are more stable when augmented with history of points. Note that prefix lengths larger than 32 doesn't perform very well in practice because they have fewer than 32 shots to propose a good point in a single trajectory. Empirically, we found prefix length equal to half of the trajectory length to perform well across the experiments. We provide more details about the ablations, experimental setup, and model hyper-parameters in the Appendix B and C.

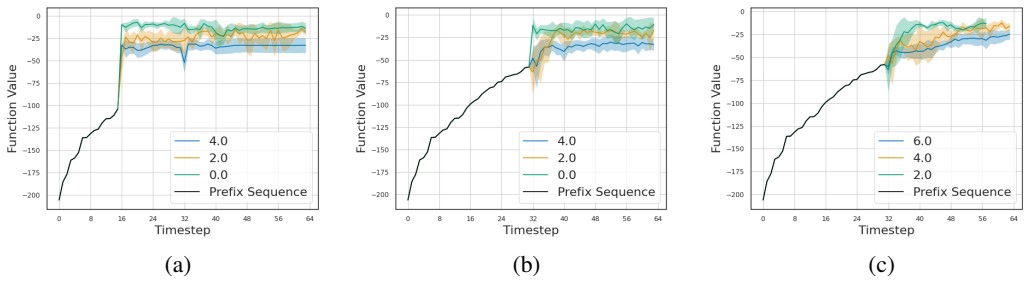

Figure 4: Rolled-out trajectories for Branin task for multiple $\hat{R}$ values (averaged over 5 runs). Figure (b) shows trajectories with prefix length 32, without updating RB (default evaluation setting). In Figure (a), we change the prefix length to 16 while in Figure (c), we update RB in the suffix.

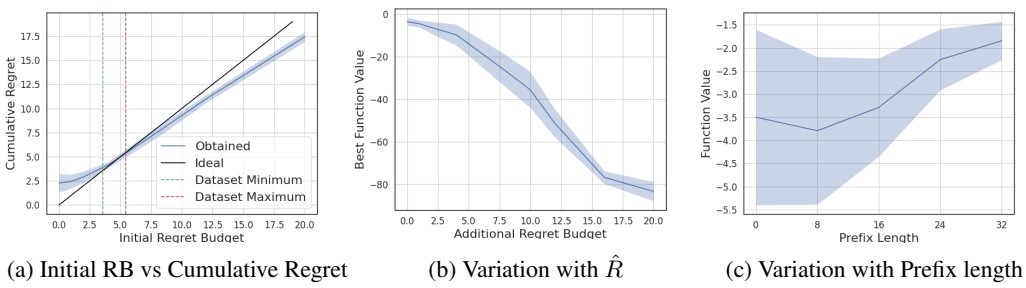

(a) Initial RB vs Cumulative Regret     (b) Variation with $\hat{R}$     (c) Variation with Prefix length

Figure 5: Ablation studies for Branin task. All the values are averaged over 5 runs.

## 3.2 DESIGN-BENCH TASKS

Next, we evaluate BONET on 7 complex real-world tasks of Design-Bench (Trabucco et al., 2022)[1]. **TF-Bind-8** and **TF-Bind-10** are discrete tasks where the goal is to optimize for a DNA sequence which has maximum affinity to bind with a particular transcription factor. The sequences are of length 8 (10) for TF-Bind-8 (TF-Bind-10), where each element in the sequence is one of 4 bases. **ChEMBL** is a discrete task where the aim is to design a drug with certain qualities. **NAS** is a discrete task where we want to optimize a NN for performance on CIFAR10 (Krizhevsky et al., 2010). In **D'Kitty** and **Ant** morphology tasks, we optimize the morphology of two robots: Ant from OpenAI gym (Brockman et al., 2016) and D'Kitty from ROBEL (Ahn et al., 2019). In **Superconductor** task, the aim is to find a chemical formula for a superconducting material with high critical temperature. D'Kitty, Ant and Superconductor are continuous tasks with dimensions 56, 60, and 86 respectively. For the first four tasks, we have query access to the exact oracle function. For Superconductor, we only have an approximate oracle, which is a random forest regressor trained on a much larger hidden dataset. These tasks are considered challenging due to high dimensionality, low quality points in the offline dataset, approximate oracles in some cases, and highly sensitive landscapes with narrow optima regions (Trabucco et al., 2022).

**Baselines** We compare BONET with multiple canonical baselines like gradient ascent, REIN-FORCE (Sutton et al., 1999), BayesOpt (Snoek et al., 2012) and CMA-ES (Hansen, 2006). We also compare with more recent methods like MINs (Kumar & Levine, 2020), COMs (Trabucco et al., 2021) and CbAS (Brookes et al., 2019).[2] For inherently active methods like BayesOpt, since we cannot query the oracle function $f(\mathbf{x})$ during optimization (due to being in an offline setting), we follow the procedure used by Trabucco et al. (2022), and perform BayesOpt on a surrogate model $\hat{f}(\mathbf{x})$ (a feedforward NN) trained on the offline dataset. For the BayesOpt baseline, we use a Gaussian Process to quantify uncertainty and use the quasi-Expected Improvement (Wilson et al., 2017)

---

[1]We haven't included Hopper since the domain is buggy - we found that the oracle function used to evaluate the task was highly inaccurate and noisy. An expanded discussion can be found in Appendix B.7

[2]We take the baseline implementations from https://github.com/brandontrabucco/design-baselines

Table 2: **100th percentile** comparative evaluation of BONET over 7 tasks averaged over 5 runs. The error bars refer to the standard deviation across the 5 seeds. We report normalized scores with $Q = 256$ (except for NAS, where we use $Q = 128$ due to compute restrictions) and highlight the top 2 results in each column. Blue denotes the best entry in the column, and Violet denotes the second best. We observe that BONET is in the top 2 in 5 out of 7 tasks, and is also consistently able to outperform the best offline dataset point in all tasks.

| BASELINE | TFBIND8 | TFBIND10 | SUPERCONDUCTOR | ANT | D'KITTY | CHEMBL | NAS | MEAN SCORE (↑) | MEAN RANK (↑) |
|---|---|---|---|---|---|---|---|---|---|
| $\mathcal{D}$ (best) | 0.439 | 0.467 | 0.399 | 0.565 | 0.884 | 0.605 | 0.436 | - | - |
| CbAS | $0.958 \pm 0.018$ | $0.657 \pm 0.017$ | $0.45 \pm 0.083$ | $0.876 \pm 0.015$ | $0.896 \pm 0.016$ | $0.640 \pm 0.005$ | $0.683 \pm 0.079$ | $0.737 \pm 0.033$ | 5.4 |
| GP-qEI | $0.824 \pm 0.086$ | $0.635 \pm 0.011$ | $0.501 \pm 0.021$ | $0.887 \pm 0.0$ | $0.896 \pm 0.0$ | $0.633 \pm 0.000$ | $1.009 \pm 0.059$ | $0.769 \pm 0.026$ | 5.3 |
| CMA-ES | $0.933 \pm 0.035$ | $0.679 \pm 0.034$ | $0.491 \pm 0.004$ | $1.436 \pm 0.928$ | $0.725 \pm 0.002$ | $0.636 \pm 0.004$ | $0.985 \pm 0.079$ | $0.840 \pm 0.156$ | 4.0 |
| Gradient Ascent | $0.981 \pm 0.015$ | $0.659 \pm 0.039$ | $0.504 \pm 0.005$ | $0.34 \pm 0.034$ | $0.906 \pm 0.017$ | $0.647 \pm 0.020$ | $0.433 \pm 0.000$ | $0.638 \pm 0.018$ | 4.0 |
| REINFORCE | $0.959 \pm 0.013$ | $0.64 \pm 0.028$ | $0.481 \pm 0.017$ | $0.261 \pm 0.042$ | $0.474 \pm 0.202$ | $0.636 \pm 0.023$ | $-1.895 \pm 0.000$ | $0.222 \pm 0.046$ | 6.7 |
| MINs | $0.938 \pm 0.047$ | $0.659 \pm 0.044$ | $0.484 \pm 0.017$ | $0.942 \pm 0.018$ | $0.944 \pm 0.009$ | $0.653 \pm 0.002$ | $0.717 \pm 0.046$ | $0.762 \pm 0.026$ | 3.7 |
| COMs | $0.964 \pm 0.02$ | $0.654 \pm 0.02$ | $0.423 \pm 0.033$ | $0.949 \pm 0.021$ | $0.948 \pm 0.006$ | $0.648 \pm 0.005$ | $0.459 \pm 0.139$ | $0.720 \pm 0.034$ | 4.4 |
| BONET (Ours) | $0.975 \pm 0.004$ | $0.681 \pm 0.035$ | $0.437 \pm 0.022$ | $0.976 \pm 0.012$ | $0.954 \pm 0.012$ | $0.654 \pm 0.019$ | $0.724 \pm 0.008$ | $0.772 \pm 0.016$ | 2.4 |

Table 3: **50th percentile** evaluations on all the tasks. Similar to Table 2, we find that BONET achieves both the best average rank and the best mean score on all tasks. This shows that BONET consistently outputs good candidate points.

| BASELINE | TFBIND8 | TFBIND10 | SUPERCONDUCTOR | ANT | D'KITTY | CHEMBL | NAS | MEAN SCORE | MEAN RANK |
|---|---|---|---|---|---|---|---|---|---|
| CbAS | $0.422 \pm 0.007$ | $0.458 \pm 0.001$ | $0.111 \pm 0.009$ | $0.384 \pm 0.010$ | $0.752 \pm 0.003$ | $0.633 \pm 0.000$ | $0.292 \pm 0.027$ | $0.436 \pm 0.008$ | 5.7 |
| GP-qEI | $0.443 \pm 0.004$ | $0.494 \pm 0.002$ | $0.299 \pm 0.002$ | $0.272 \pm 0.006$ | $0.754 \pm 0.004$ | $0.633 \pm 0.000$ | $0.544 \pm 0.099$ | $0.491 \pm 0.016$ | 4.0 |
| CMA-ES | $0.543 \pm 0.007$ | $0.483 \pm 0.011$ | $0.376 \pm 0.004$ | $0.685 \pm 0.018$ | $0.637 \pm 0.148$ | $0.633 \pm 0.000$ | $0.591 \pm 0.102$ | $0.466 \pm 0.020$ | 3.4 |
| Gradient Ascent | $0.572 \pm 0.024$ | $0.470 \pm 0.004$ | $0.463 \pm 0.022$ | $0.141 \pm 0.010$ | $0.637 \pm 0.148$ | $0.633 \pm 0.000$ | $0.433 \pm 0.000$ | $0.478 \pm 0.029$ | 4.3 |
| REINFORCE | $0.450 \pm 0.003$ | $0.472 \pm 0.000$ | $0.470 \pm 0.017$ | $0.146 \pm 0.009$ | $0.307 \pm 0.002$ | $0.633 \pm 0.000$ | $-1.895 \pm 0.000$ | $0.083 \pm 0.004$ | 4.9 |
| MINs | $0.425 \pm 0.011$ | $0.471 \pm 0.004$ | $0.330 \pm 0.011$ | $0.651 \pm 0.010$ | $0.890 \pm 0.003$ | $0.633 \pm 0.000$ | $0.433 \pm 0.000$ | $0.547 \pm 0.005$ | 4.0 |
| COMs | $0.492 \pm 0.009$ | $0.472 \pm 0.012$ | $0.365 \pm 0.026$ | $0.525 \pm 0.018$ | $0.885 \pm 0.002$ | $0.633 \pm 0.000$ | $0.287 \pm 0.173$ | $0.522 \pm 0.034$ | 3.8 |
| BONET | $0.505 \pm 0.055$ | $0.496 \pm 0.037$ | $0.369 \pm 0.015$ | $0.819 \pm 0.032$ | $0.907 \pm 0.020$ | $0.630 \pm 0.000$ | $0.571 \pm 0.095$ | $0.614 \pm 0.035$ | 2.8 |

acquisition function for optimization, similar to prior work (Trabucco et al., 2022; 2021). MINs (Kumar & Levine, 2020) also train a forward model to optimize over the conditioning parameters. Our method does not need a separate forward model and thus, is not dependent on the quality of the fit of the forward model. We provide other variants of BayesOpt baseline in Appendix D.2.

**Evaluation** We allow a query budget of $Q = 256$ for all the baselines, except for NAS, where we use a reduced budget of $Q = 128$ due to compute restrictions. For BONET, across all tasks, we roll out 4 trajectories, each with a prefix and prediction subsequence length of 64 each. The prediction subsequence is initialized with one of 4 candidate low $\hat{R}$ values $0, 0.01, 0.05, 0.1$. For each $\hat{R}$, we then roll out for 64 timesteps and choose the best point. We report the mean and standard deviation over 5 trials for each of the models and tasks in Table 2. Following the procedure used by (Trabucco et al., 2021; 2022; Yu et al., 2021), the results of Table 2 are linearly normalized between the minimum and maximum values of a large hidden offline dataset. In Table 3, we also present the median function value of the the proposed output points for each method, averaged over 5 runs.

**Results** Overall, we obtain a mean score of $0.772$ and an average rank of $2.4$, which is the best among all the baselines. We also achieve the best results on three tasks. Additionally, we are among the top 2 for five out of seven tasks. We show significant improvements over generative methods such as MINs (Kumar & Levine, 2020) or CbAS (Brookes et al., 2019), and forward mapping methods such as COMs (Trabucco et al., 2021) on TF-Bind-8, TF-Bind-10, Ant and D'Kitty. We also note that while BONET is placed second in Ant, it shows a much lower standard deviation ($0.012$) compared to the to the best performing method CMA-ES, which has a much larger standard deviation of $0.928$. We also have a lowest mean standard deviation across all tasks, suggesting that BONET is less sensitive to bad initializations compared to other methods. We report the unnormalized results, ablations and other experimental details for Design-Bench tasks in Appendix C. BONET also performs best on $50$th percentile evaluation, showing that it has a better set of proposed points compared to other methods, and that the performance is not by randomly finding good points.

## 4  RELATED WORK

**Active BBO** The majority of prior work in BBO have been in the active setting, where surrogate models are allowed to query the function during training. This includes long bodies of work

in Bayesian Optimization, e.g., (Snoek et al., 2012; Swersky et al., 2013; Srinivas et al., 2010; Nguyen & Osborne, 2020) and bandits, e.g., (Garivier et al., 2016; Riquelme et al., 2018; Joachims et al., 2018b;a; Swaminathan & Joachims, 2015; Jacq et al., 2019). Such methods usually employ surrogate models such as Gaussian Processes (Srinivas et al., 2010), Neural Processes (Garnelo et al., 2018a;b; Gordon et al., 2019; Anonymous, 2022; Singh et al., 2019) or Bayesian Neural Networks (Chang, 2021; Goan & Fookes, 2020) to approximate the black-box function, and an uncertainty-aware acquisition strategy for querying new points.

**Offline BBO** Recent works have made use of such datasets and shown promising results (Kumar & Levine, 2020; Trabucco et al., 2021; Brookes et al., 2019; Fannjiang & Listgarten, 2020; Fu & Levine, 2021; Yu et al., 2021). Kumar & Levine (2020) train a stochastic inverse mapping from the outputs $y$ to inputs $\mathbf{x}$ using a generative model similar to a conditional GAN (Goodfellow et al., 2014; Mirza & Osindero, 2014; Arjovsky et al., 2017; Nowozin et al., 2016). They then optimize over $y$ to find a good design point. Training GANs however can be difficult due to issues like mode collapse (Arjovsky et al., 2017). Other methods make use of gradient ascent to find an optimal solution. Trabucco et al. (2021) and Yu et al. (2021) train a model to be robust to outliers by regularizing the objective such that they assign low score to those points. Fu & Levine (2021) train a normalized maximum likelihood estimate of the function. We instead offer a fresh perspective based on generative sequence modelling and we show strong results in comparison to many of these prior works in Section 3.

**Offline reinforcement learning (RL)** While both RL and BBO are sequential decision-making problems, the key difference is that RL is stateful while BBO is not. In the offline setting (Schmidhuber, 2019; Jacq et al., 2019), both problems require models or policies that can generalize beyond the offline dataset to achieve good performance. Related to our work, autoregressive transformers have been successfully applied on trajectory data obtained from offline RL (Chen et al., 2021; Janner et al., 2021). However, there are important differences between their setting and offline BBO. For example, the data in offline BBO is not sequential in nature, unlike in offline RL where the offline data is naturally in the form of demonstrations. One of our contributions is to design a notion of 'high-to-low' sequences to BBO for autoregressive modeling and test-time generalization.

**Learned optimizers** Our work also bears resemblance to work on meta-learning optimizers like those of (Andrychowicz et al., 2016) and (Chen et al., 2017). However, a key difference between these learned optimizers and BONET is that they require access to gradients either during training time or during both training and evaluation time, whereas BONET has no such restriction, meaning it can work in situations where access to gradient information is not practical (for instance with non-differentiable black-box functions). Furthermore, we concentrate on the offline setting, in contrast to learned optimizer work which usually looks at an active optimization setting.

## 5 DISCUSSION

We presented BONET, a novel generative framework for pretraining black-box optimizers using offline data. BONET consists of a 3 phased process. In the first phase, we use a novel SORT-SAMPLE strategy to generate trajectories from offline data that use the sorting heuristic to mimic the behavior of online BBO optimizers. In phases 2 and 3, we train our model using an autoregressive transformer and use it to generate candidate points that maximize the black-box function. Experimentally, we verify that BONET is capable of solving complex high-dimensional tasks like the ones in Design-Bench, achieving an average rank of **2.4** with a mean score of **0.772**.

**Limitations and Future Work** BONET assumes knowledge of the approximate value of the optima $f(\mathbf{x}^*)$ to computes the regrets of points. Though we show in Appendix C.5 that different reasonable estimates of $f(\mathbf{x}^*)$ can give similar results, this is still something we would like to address in future work. We are also interested in extending BONET to an active setting where our model can quantify uncertainty and actively query the black-box function after pretraining on an offline dataset. On a practical side, we would also be interested in analyzing the properties of domains that dictate where BONET can strongly excel (e.g., D'Kitty) or struggle (e.g., Superconductor) relative to other approaches.Finally, we aim to expand our scope to a meta-task setting, where instead of a single function, our offline data consists of past evaluations from multiple black-box functions.

## 6 ETHICAL AND SOCIAL RESPONSIBILITY

Offline black-box optimization can play a critical role in improving efficiency and safety in many real-world settings like nuclear reactors or the pharmaceutical industry, where active optimization may prove to be computationally inefficient (requiring too many queries) or even dangerous. However, while we do not anticipate anything inherently malicious about our work, it is possible to utilize our proposed method (and other optimizers in general) in malicious settings (e.g., optimizing for drugs that have harmful effects). This is something to remember when deploying algorithms such as ours for high stake real-world applications.

## 7 REPRODUCIBILITY

Throughout the paper, we maintained a high standard of rigor for reproducibility. We report the detailed pseudocode of our method in Algorithm 1, and we also provide the link to our code via an anonymized link here. We provide more details on our training setup and choice of hyperparameters in Appendix B. We report results on multiple datasets in Design-Bench (Trabucco et al., 2022), each with different properties and benchmark our method over multiple baselines from different families of approaches. Our results are averaged over 5 seeds, and we also provide the standard deviations. We also conduct several ablations to evaluate sensitivity of BONET to different parameters.

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

# A  NOTATIONS & THEORETICAL ANALYSIS

## A.1  NOTATIONS

For ease of reference, we list the notations in Table 4.

Table 4: Important notations used in the paper

| SYMBOL | MEANING |
|---|---|
| $f$ | Black box function |
| $\mathcal{X}$ | Support of $f$ |
| $\mathbf{x}^*$ | Optima (taken to be maxima for consistency) |
| $\mathcal{D}$ | Offline dataset |
| $N$ | Size of offline dataset |
| $\mathcal{D}_{\text{traj}}$ | Trajectory dataset |
| num_trajs | Number of trajectories in $\mathcal{D}_{\text{traj}}$ |
| $\mathcal{T}$ | A trajectory |
| $T$ | Length of a trajectory |
| $Q$ | Query budget for black-box function |
| $P$ | Prefix length |
| $R_i$ | Regret Budget at timestep $i$ |
| $R_1$ | Initial Regret Budget |
| $\hat{R}$ | Evaluation Regret Budget |
| $g_\theta$ | Autoregressive generative model with parameters $\theta$ |
| $K, \tau$ | SORT-SAMPLE hyperparameters |
| $N_B$ | Number of bins |
| $C$ | Context Length |

## A.2  THEORETICAL ANALYSIS

One crucial component in the SORT-SAMPLE is sorting the trajectories in the increasing order of the function values. Although our primary motivation for sorting is derived from the empirical observations from online black-box optimizers (Figure 1c), we note that for a certain class of functions that are non-trivial to solve from the perspective of optimization (maximization for our paper), lower (higher) function values occupy a larger (smaller) domain. Thus, intuitively we can relate lower function values with exploration and higher function values with exploitation. With this perspective, sorting can be seen as moving from a high-diversity region to a low-diversity region - a behavior typically seen in online black-box optimizers (Bijl et al., 2016; Garivier et al., 2016). In this section, we try to formally prove such properties for this constrained class of functions.

We consider the simplified case of differential 1D functions with certain assumptions for simplicity, and further extend this notion to a more general $D$-dimensional case. First, we define a notion of $\epsilon$-high points.

**Definition 1.** $\epsilon$-high values. *Let the range of $f$ be denoted as $[y_{min}, y_{max}]$. Then, a function value $y$ in this range is $\epsilon$-high if $y \geq y_{min} + \epsilon(y_{max} - y_{min})$.*

Intuitively, the above definition implies that $y$ is $\epsilon$-high if it is in the top $1 - \epsilon$ fraction of the range of $f$. The following result characterizes the relative diversity of regions consisting of $\epsilon$-high points for 1-D functions.

**Proposition 1.** *Let $f : [a, b] \to \mathbb{R}$, $a, b \in \mathbb{R}$, be a real-valued, continuous and differentiable function such that the $f(a)$ and $f(b)$ are not $\epsilon$-high. Let $H \subseteq [a, b]$ be a Lebesgue-measurable set of $x$ values for which $f(x)$ is $\epsilon$-high, with $\epsilon > 0.5$. Let $H^c = [a, b] \setminus H$. Without loss of generality, let's assume that $f(a) \leq f(b)$. If the Lipchitz constant $L$ of $f$ is upper bounded by*

$$\frac{2(\epsilon(y_{max} - y_{min}) + y_{min} - f(a))}{b - a},$$ (6)

*then $|H| < |H^c|$, where $|\cdot|$ denotes volume of a set w.r.t. the Lebesgue Measure.*

**Proof of Proposition 1.**

Note that if no point in the domain $[a, b]$ achieves $\epsilon$-high function value, then the statement holds trivially true. So, we assume that there is atleast one point which has $\epsilon$-high function value. Let $x_1$ and $x_2$ be the smallest and largest such points in the domain. Since the boundary points doesn't have $\epsilon$-high values, $|H^c| \geq (x_1 - a) + (b - x_2)$ and $|H| \leq (x_2 - x_1)$. Thus, if we prove that $(x_1 - a) + (b - x_2) \geq (x_2 - x_1)$, then we are done. To prove this, we try to prove $(x_1 - a) \geq (x_2 - x_1)$.

Assume, on the contrary, that $(x_1 - a) < (x_2 - x_1)$. Rearranging, we get

$$\frac{2}{x_2 - a} < \frac{1}{x_1 - a} \tag{7}$$

Now, by the definition of Lipchitz constant, we have:

$$\frac{f(x_1) - f(a)}{x_1 - a} \leq L$$
$$\Rightarrow \frac{2(f(x_1) - f(a))}{x_2 - a} \overset{7}{\leq} L \tag{8}$$
$$\Rightarrow \frac{2(\epsilon * (y_{max} - y_{min}) + y_{min} - f(a))}{b - a} \leq L$$

Last inequality holds because $x_2 - a \leq b - a$. This inequality contradicts the bound 6 on $L$, completing our proof for Proposition 1. □

Now, we show an extension of this proposition for $D$-dimensional functions with hypercube domains.

**Proposition 2.** *Let $f : \mathcal{X} \to \mathbb{R}$ be a $D$-dimensional, real-valued, continuous, and differentiable function with hypercube domain $\mathcal{X} = [a_1, b_1] \times [a_2, b_2], \cdots, \times[a_D, b_D]$ , such that none of the boundary points are $\epsilon$-high, for some fixed $\epsilon$. Here by boundary points, we mean the points on the surface of the domain hypercube. Let $y_{max}$ and $y_{min}$ be the maximum and the minimum values attained by $f$. Let $H \subseteq \mathcal{X}$ be a Lebesgue-measurable set of points for which $f(\mathbf{x})$ is $\epsilon$-high. Let $H^c = \mathcal{X} \setminus H$. If the Lipchitz constant $L$ of $f$ is upper bounded by*

$$\frac{2(\epsilon * (y_{max} - y_{min}) + y_{min} - \max_{x_2, \cdots, x_D} f(a_1, x_2, \cdots, x_D))}{b_1 - a_1}, \tag{9}$$

*then $|H| < |H^c|$, where $|\cdot|$ denotes volume of a set w.r.t. the Lebesgue Measure.*

**Proof.** We prove this proposition by induction on the number of dimensions $D$. Notice that for $D = 1$, the statement reduces to Proposition 1, which we have already proved. Next, we assume that the statement holds for $(D - 1)$-dimensional functions and prove it for $D$ dimensions, with $D > 1$.

Let's define $\mathcal{H}_{D,\epsilon} : \mathcal{F}_D \to \mathcal{B}_D$ to be a functional that maps any $D$-dimensional function, say $f$, to a Lebesgue-measurable subset of $\mathbb{R}^D$ that corresponds to the set of points where $f(\mathbf{x})$ is $\epsilon$-high. , Here, $\mathcal{F}_D$ and $\mathcal{B}_D$ denote the set of all $D$-dimensional functions and the set of all Lebesgue-measurable subsets of $\mathbb{R}^D$ respectively.

We similarly define the mapping $\mathcal{H}_{D,\epsilon}^c$ to be a functional mapping a function $f$ to the complement of its $\epsilon$-high region. Thus, $H = \mathcal{H}_{D,\epsilon}(f)$ and $H^c = \mathcal{H}_{D,\epsilon}^c(f)$. Now, by definition,

$$|\mathcal{H}_{D,\epsilon}(f(\bullet, \cdots, \bullet)| = \int_{x_D} |\mathcal{H}_{D-1,\epsilon}(f(\bullet, \cdots, \bullet, x_D))| \, dx_D \tag{10}$$

And similarly,

$$|\mathcal{H}_{D,\epsilon}^c(f(\bullet, \cdots, \bullet)| = \int_{x_D} |\mathcal{H}_{D-1,\epsilon}^c(f(\bullet, \cdots, \bullet, x_D))| \, dx_D \tag{11}$$

Consequently, to prove $|\mathcal{H}_{D,\epsilon}(f(\bullet, \cdots, \bullet)| \leq |\mathcal{H}_{D,\epsilon}^c(f(\bullet, \cdots, \bullet)|$, we prove $|\mathcal{H}_{D-1,\epsilon}(f(\bullet, \cdots, \bullet, x_D))| \leq |\mathcal{H}_{D-1,\epsilon}^c(f(\bullet, \cdots, \bullet, x_D))|$ for every $x_D \in [a_D, b_D]$.

To do this, we first fix the $D^{th}$ dimension to be $c$. In other words, we are considering the $(D-1)$-dimensional slice of $f(x_1, \cdots, x_D)$ with $x_D = c$. Let $g$ be such a slice with $g(x_1, \cdots, x_{D-1}) = f(x_1, \cdots, x_{D-1}, c)$. First we need $\epsilon^g$ for which $\epsilon^g$-high value for $g$ is $\epsilon$-high for $f$:

$$\epsilon^g(y_{max}^g - y_{min}^g) + y_{min}^g = \epsilon(y_{max} - y_{min}) + y_{min} \tag{12}$$

where $y_{max}^g$ and $y_{min}^g$ are the minimum and maximum values respectively achieved by $g$. By this choice of $\epsilon^g$, we are ensuring that a point $(x_1, \cdots, x_{D-1}, c)$ is $\epsilon$-high w.r.t $f$ if and only if $(x_1, \cdots, x_{D-1})$ is $\epsilon^g$-high w.r.t $g$. In other words,

$$\begin{aligned}
\mathcal{H}_{D-1,\epsilon^g}(g) &= \mathcal{H}_{D-1,\epsilon}(f(\bullet, \cdots, \bullet, c)) \\
\mathcal{H}_{D-1,\epsilon^g}^c(g) &= \mathcal{H}_{D-1,\epsilon}^c(f(\bullet, \cdots, \bullet, c))
\end{aligned} \tag{13}$$

Let the Lipchitz constant of $g$ be $L^g$. First we show that $L_g \leq L$. By definition of Lipchitz constant, for $\mathbf{x} = (x_1, \cdots x_{D-1}), \mathbf{z} = (z_1, \cdots, z_{D-1})$ in the domain of $g$,

$$\begin{aligned}
L_g &= \sup_{\mathbf{x} \neq \mathbf{z}} \frac{|g(x_1, \cdots, x_{D-1}) - g(z_1, \cdots, z_{D-1})|}{\sqrt{\sum_{i=1}^{D-1}(x_i - z_i)^2}} \\
&= \sup_{\mathbf{x} \neq \mathbf{z}} \frac{|f(x_1, \cdots, x_{D-1}, c) - f(z_1, \cdots, z_{D-1}, c)|}{\sqrt{\sum_{i=1}^{D-1}(x_i - z_i)^2 + (c-c)^2}} \\
&\leq L
\end{aligned} \tag{14}$$

Where last inequality is by definition of $L$ w.r.t $f$. Combining this with our bound on $L$ in 9, we get

$$\begin{aligned}
L_g &\leq \frac{2(\epsilon * (y_{max} - y_{min}) + y_{min} - \max\limits_{x_2, \cdots, x_D} f(a_1, x_2, \cdots, x_D))}{b_1 - a_1} \\
&\leq \frac{2(\epsilon * (y_{max} - y_{min}) + y_{min} - \max\limits_{x_2, \cdots, x_{D-1}} f(a_1, x_2, \cdots, c))}{b_1 - a_1} \quad \text{(fixing } D^{th} \text{ dimension)} \\
&\overset{12}{\leq} \frac{2(\epsilon^g * (y_{max}^g - y_{min}^g) + y_{min}^g - \max\limits_{x_2, \cdots, x_{D-1}} g(a_1, x_2, \cdots, x_{D-1}))}{b_1 - a_1}
\end{aligned} \tag{15}$$

Thus, the Lipchitz bound assumption is followed by $g$ with $\epsilon = \epsilon^g$. Also, the boundaries are not $\epsilon^g$-high w.r.t $g$ because of the choice of $\epsilon^g$. This implies, by inductive assumption, that $|\mathcal{H}_{D-1,\epsilon^g}(g)| \leq |\mathcal{H}_{D-1,\epsilon^g}^c(g)|$. This, combined with equality 13 proves that that $|\mathcal{H}_{D-1,\epsilon}(f(\bullet, \cdots, \bullet, c))| \leq |\mathcal{H}_{D-1,\epsilon}^c(f(\bullet, \cdots, \bullet, c))|$. Since this is true for all $c \in [a_D, b_D]$, by equations 10 and 11, our proof for $|\mathcal{H}_{D,\epsilon}(f(\bullet, \cdots, \bullet))| \leq |\mathcal{H}_{D,\epsilon}^c(f(\bullet, \cdots, \bullet))|$ is complete.

$\square$

# B  EXPERIMENTAL DETAILS

## B.1  SORT-SAMPLE

In SORT-SAMPLE, the score of each bin is calculated according to the formula

$$s_i = \frac{|B_i|}{|B_i| + K} \exp\left(\frac{-|\hat{y} - y_{b_i}|}{\tau}\right)$$

The two variables $K$ and $\tau$ here act as smoothing parameters. $K$ controls the relative priority given to the larger bins (bins with more points). Higher value of $K$ assigns higher relative weight to these large bins compared to smaller bins, whereas a low value of $K$ the weight assigned to large and small bins would be similar. In the extreme case where $K = 0$, the weight due to $\frac{|B_i|}{|B_i| + K}$ will always be 1, regardless of bin size. For very large value of $K$, the weight will be approximately linearly proportional to the bin size $|B_i|$. The later case is not desirable because if there is a bin

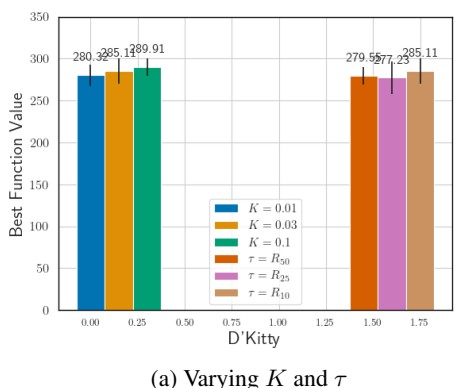
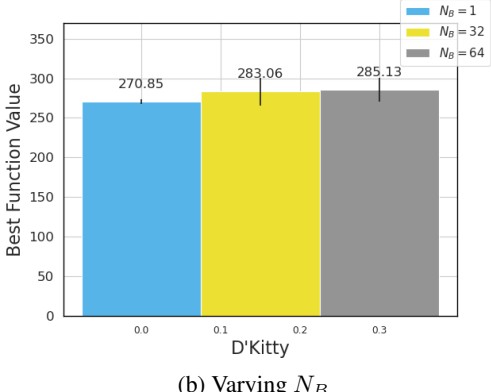

(a) Varying $K$ and $\tau$          (b) Varying $N_B$

Figure 6: We plot the best achieved function value for different combinations of $K$, $\tau$ and $N_B$. As expected, we don't see much sensitivity to the choice of $K$ and $\tau$. For $N_B$, as expected, we a significant decrease for $N_B = 1$, but $N_B = 32$ is comparable to $N_B = 64$

which has very large number of points (which might be a low quality bin), then most of the total weight will be given to that bin because of the linear proportionality.

Temperature $\tau$ controls how harshly the bad bins are penalized. Lower the $\tau$ value, lower the relative score of the low quality bins (bins with high regret) and vice versa.

In our experiments, we don't tune the values of $K$, $\tau$, and number of bins $N_B$. In all the tasks, we use $K = 0.03 \times N$ and $\tau = R_{10}$, where $R_{10}$ is the $10^{th}$ percentile regret value in $\mathcal{D}$. For Branin task, we use , $N_B = 32$ and for all the Design-Bench tasks, we use $N_B = 64$. Empirically, as we show in Figure 6a, we didn't observe much effect on $K$ in the range $[0.01, 0.1]$, and for $\tau$ from the $50^{th}$ to the $10^{th}$ percentiles of $R$. Figure 6b shows variation with $N_B$, keeping all other parameters fixed. As expected, $N_B = 1$ doesn't perform well, as having just one bin is equivalent to having no re-weighting. Beyond 32, we don't see much variation with the value of $N_B$.

## B.2 MODEL ARCHITECTURE & IMPLEMENTATION

**Architecture** We use a GPT (Radford et al., 2019) like architecture, where each timestep refers to two tokens $R_t$ and $\mathbf{x}_t$. Similar to Chen et al. (2021), we add a new learned timestep embedding (in addition to the positional embedding already present in transformers). Each token $R_t$ and $\mathbf{x}_t$ that goes as input to our model is first projected into a 128 dimensional embedding space using a linear embedding layer. To this embedding, we also add the positional and timestep embeddings. This is passed through a causally masked transformer. The prediction head for $R_t$ predicts $\hat{\mathbf{x}}_t$, which is then used to compute the loss. The output of the prediction head for $\mathbf{x}_t$ is discarded. At each timestep, we feed in the last $C$ timesteps to the model, where $C$ here refers to the context length. For continuous tasks, the prediction head for $R_t$ outputs a $d$-dimensional prediction $\hat{\mathbf{x}}_t$. For discrete tasks, the prediction head outputs a $V \times d$-dimensional prediction, where $V$ refers to the number of classes in the discrete task. Thus, each dimension in $\mathcal{X}$ corresponds to a $V$-dimensional logits vector.

**Code** Our code (available at the anonymized link here) is built upon the code from minGPT [3] and Chen et al. (2021) [4]. All code we use is under the MIT licence.

**Training** The parameter details for all the tasks are summarized in the Table 5. Note that almost all of the parameters are same across all the Design-Bench tasks. Number of layers is higher for continuous tasks, as they are of higher dimensionalities. For all the tasks, we use a batch size of 128 and a fixed learning rate of $10^{-4}$ for 75 epochs. All training is done using 10 Intel(R) Xeon(R) CPU cores (E5-2698 v4443 @ 2.20GHz) and one NVIDIA Tesla V100 SXM2 GPU.

---

[3] https://github.com/karpathy/minGPT
[4] https://github.com/kzl/decision-transformer

## B.3 EVALUATION

For all the Design-Bench tasks (except NAS), we use a query budget $Q = 256$. For NAS, we use a query budget of $Q = 128$ due to compute restrictions. Since we use a trajectory length of $128$ and a prefix length of $64$, this means that we can roll-out four different trajectories. There are two variable parameters during the evaluation: Evaluation RB ($\hat{R}$) and the prefix sub-sequence. We empirically observed that $\hat{R}$ has more impact on the variability of rolled-out points compared to prefix sub-sequence. Hence, we roll-out trajectory for $4$ different low $\hat{R}$ values $(0.0, 0.01, 0.05, 0.1)$. These values are kept fixed across all the tasks and are not tuned. They are chosen to probe the interval $[0.0, 0.1]$, while giving slightly more importance to the low values by choosing $0.01$. Evaluation strategies with lower query budget $Q$ available are discussed in the section C.2

Table 5: Important parameters for all the tasks

| TASK | Type | HEADS | LAYERS | $T$ | $P$ | $C$ | $N_B$ | num_trajs |
|---|---|---|---|---|---|---|---|---|
| Branin (toy) | Continuous | 4 | 8 | 64 | 32 | 32 | 32 | 400 |
| TFBind8 | Discrete | 8 | 8 | 128 | 64 | 64 | 64 | 800 |
| TFBind10 | Discrete | 8 | 8 | 128 | 64 | 64 | 64 | 800 |
| ChEMBL | Discrete | 8 | 8 | 128 | 64 | 64 | 64 | 800 |
| NAS | Discrete | 8 | 8 | 128 | 64 | 64 | 64 | 800 |
| D'Kitty | Continuous | 8 | 32 | 128 | 64 | 64 | 64 | 800 |
| Ant | Continuous | 8 | 32 | 128 | 64 | 64 | 64 | 800 |
| Superconductor | Continuous | 8 | 32 | 128 | 64 | 64 | 64 | 800 |

## B.4 FIXED VS. UPDATED RB DURING EVALUATION

During the evaluation of the prediction sequence, we proposed to keep the Regret Budget (RB) fixed. One alternative strategy is to sequentially update RB after every iteration, i.e. predict $\mathbf{x}_t$ from $R_t$, and compute $R_{t+1} = R_t - (f(\mathbf{x}^*) - f(\mathbf{x}_t))$. However, there are two issues with such an update rule:

1. Updating RB adds a sequential dependency on our model during evaluation, as we must query $f(\mathbf{x}_t)$ to compute $R_{t+1}$. Thus, generating the $Q$ candidate points is not purely offline.

2. While updating the regret budget $R_t$, it is possible that at some timestep $t$, $R_t$ becomes negative. Since the model has never seen negative RB values during training, this is undesirable.

Hence, we do not update RB during evaluation, and instead provide a fixed $\hat{R}$ value at every timestep after the prefix length. This way, point proposal is not dependent on sequential evaluations of $f$, making it much faster as the evaluations on $f$ can then be parallelized. Furthermore, by not updating $\hat{R}$ we sidestep the issue of negative RBs. Empirically, as we see in Figure 7, there is not much difference across different strategies, which justifies our choice of not updating RB, allowing our method to be purely offline.

## B.5 BASELINES

For the gradient ascent baseline of Branin task, we train a 2 layer neural network (with hidden layer of size $128$) as a forward model for $75$ epochs with a fixed learning rate of $10^{-4}$. For gradient ascent on the learnt model during evaluation, we report results with a step size of $0.1$ for $64$ steps. We average over 5 seeds, and for each seed we perform two random restarts.

For the baselines in the Design-Bench tasks, we run the baseline code [5] provided in Trabucco et al. (2022) and report results with the default parameters for a query budget of 256.

---

[5]We were not able to reproduce the results of (Fu & Levine, 2021) and (Yu et al., 2021) on the latest version of Design-Bench.

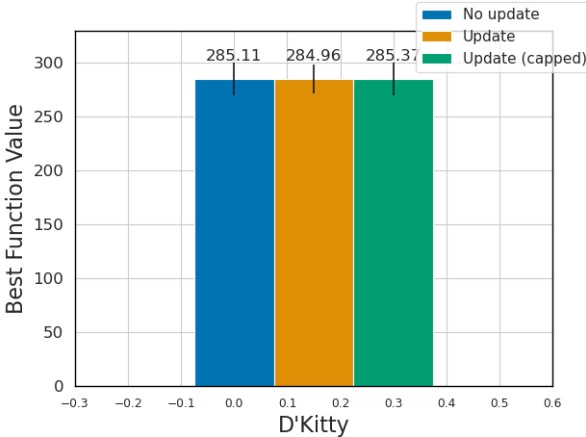

Figure 7: Comparison between 3 strategies: (1) do not update the RB at all (blue), (2) update the RB but do not handle the case when $R_t$ becomes negative (orange) and (3) don't make the update if the update is going to make the RB negative and update otherwise (green). In all three cases, we see similar performance.

## B.6 DESIGN-BENCH TASKS

For the Design-Bench tasks, we pre-process the offline dataset to normalize the function values before constructing the trajectory dataset $\mathcal{D}_{\text{traj}}$. Normalized $y_{norm}$ values are computed as $y_{norm} = \frac{y - y_{min}}{y_{max} - y_{min}}$, where $y_{min}$ and $y_{max}$ are minimum and maximum function values of a much larger hidden dataset. Note that we only use the knowledge of $y_{min}$ and $y_{max}$ from this hidden dataset, and not the corresponding $\mathbf{x}$ values. With this normalization procedure, we use 1.0 as an estimate of $f(\mathbf{x}^*)$ while constructing trajectories in $\mathcal{D}_{\text{traj}}$. The results we report in Table 2 are also normalized using the same procedure, similar to prior works (Trabucco et al., 2022; 2021). We also report unnormalized results in Table 7.

Table 6: Task details

| TASK | SIZE | DIMENSIONS | TASK MAX |
|---|---|---|---|
| TFBind8 | 32898 | 8 | 1.0 |
| TFBind10 | 10000 | 10 | 2.128 |
| ChEMBL | 1093 | 31 | 443000.0 |
| NAS | 1771 | 64 | 69.63 |
| D'Kitty | 10004 | 56 | 340.0 |
| Ant | 10004 | 60 | 590.0 |
| Superconductor | 17014 | 86 | 185.0 |

## B.7 HOPPER TASK

We didn't include the Hopper task in our results in Table 2 because of the inconsistency between the offline dataset values and the oracle outputs. Hopper data consist of 3200 points, each with 5126 dimensions. The lowest and highest function values are 87.93 and 1361.61. Figure 8 shows the distribution of the normalized function values. This distribution is extremely skewed towards low function values. Only 6 points out of 3200 have a normalized function value greater than 0.5.

We noticed that the oracle of Hopper is highly inaccurate for points with higher function values. Figure 9 shows the function values in the dataset vs. the oracle output for the top 10 best points in the data, clearly showing the inconsistency between the two. In fact, the oracle minima and maxima for the dataset are just 56.26 and 786.79, respectively, far from the actual dataset values. Due to such discrepancies, we have decided not to include the Hopper task in our analysis.

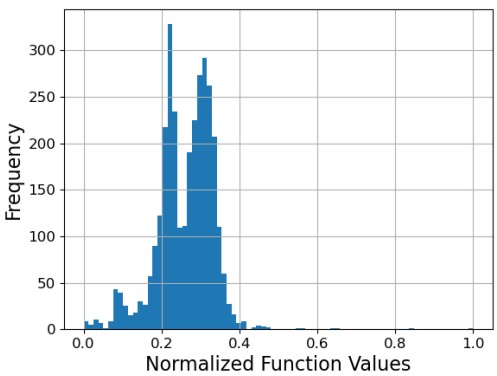 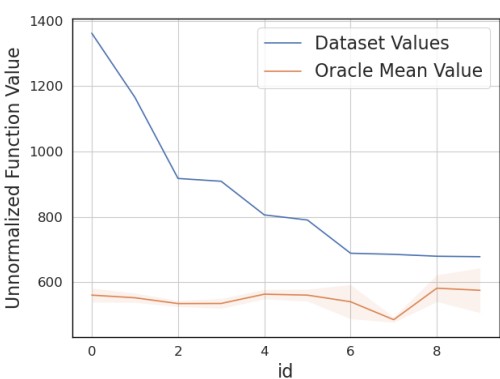

Figure 8: Histogram of normalized function values in the Hopper dataset. The distribution is highly skewed towards low function values.

Figure 9: Dataset values vs Oracle values for top 10 points. Oracle being noisy, we show mean and standard deviation over 20 runs.

## C    ADDITIONAL EXPERIMENTAL RESULTS

### C.1    ABLATIONS ON SORT-SAMPLE STRATEGY

SORT-SAMPLE algorithm has two main components: Sampling after re-weighting and sorting. Our sorting heuristic is primarily motivated by typical runs of online optimizers. To show this, we run an online GP to optimize the three synthetic functions, namely the negative Branin, negative Goldstein-Price and negative Six hump camel functions and plot the function values for the proposed points. Figure 10 shows sample trajectories of the function values of the proposed maxima after each function evaluation. We can see, on average, the function values tend to increase over time as the number of queries increases. Such behavior has also been reported for other black-box functions and setups, see e.g. (Bijl et al., 2016).

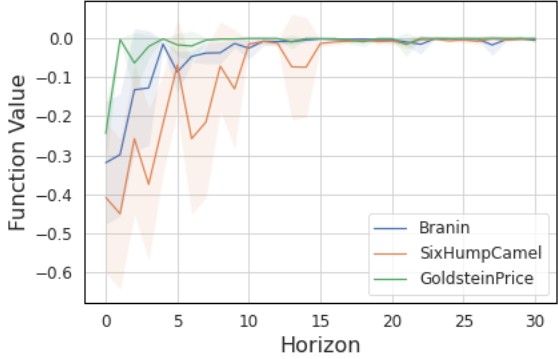

Figure 10: Mean and standard deviations of 10 trajectories unrolled by a simple GP-based BayesOpt algorithm on the 3 synthetic functions.

However, in this section, we do perform ablations to see the effects of these components. To this end, we construct trajectories using 4 strategies:

1. Random: Uniformly randomly sample a trajectory from the offline dataset.
2. Random + Sorting: Uniformly randomly sample a trajectory from the offline dataset and sort it in ascending order of the function values.
3. Re-weighting + Partial Sorting: Perform re-weighting, uniformly randomly sample $n_i$ number of points from each bin, and concatenate them from lowest quality bin to the high-

est quality bin. This way, the trajectory will be partially sorted, i.e. the order of the bins themselves is sorted, but the points sampled from a bin will be randomly ordered. In this case, the trajectories are not entirely monotonic w.r.t. the function values. Intuitively, this intermingles exploration and exploitation phases within and across bins respectively.

4. Re-weighting + Sorting (default in BONET): Sort the trajectory obtained in strategy 3. This is the default setting we use in our experiments.

Figure 11 contains the results obtained by each of the four strategies. Note that while going from strategy 1 to 2, we keep the points sampled in a trajectory the same, so the only difference between them is sorting. Figure 11 shows that strategy 1 clearly outperforms strategy 2. This means that sorting has a significant impact on the results. Next, note that strategy 2 and 4 differ only in their sampling strategy, and strategy 4 outperforms strategy 2, which shows the effectiveness of re-weighting. This experiment justifies our choice for both re-weighting and sorting.

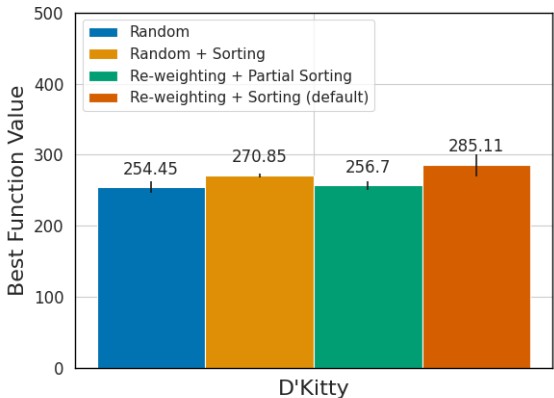

Figure 11: Results with various trajectory construction strategies for D'Kitty task, averaged over 5 runs. Comparing blue and orange bars, it is evident that sorting is improving the results. Similarly, comparison of orange bar with red bar shows that re-weighting further improves the results.

### C.2 ANALYSIS ON QUERY BUDGET $Q$

So far, we have been discussing the results with query budget $Q = 256$. Here, we describe the evaluation strategy we use when lower query budget is available. Our strategy will be to give higher preference to lower $\hat{R}$ values when lower budget is available. For example, when $Q = 192$, we only roll-out and evaluate for $\hat{R} \in \{0.0, 0.01, 0.05\}$. For $192 < Q \leq 256$, we will roll-out for $\hat{R} \in \{0.0, 0.01, 0.05, 0.1\}$, evaluate the entire predicted sub-sequences of lengths 64 for $\{0.0, 0.01, 0.05\}$, and evaluate the first $Q - 192$ points in the predicted sub-sequences for $\hat{R} = 0.1$. In the Figure 12, we present the results for different query budgets for our method compared to important baselines, for the D'Kitty task. We outperform the baselines for almost all the query budget values.

### C.3 ADDITIONAL ABLATIONS

Here we present ablations similar to Section 3 on the D'Kitty task, and observe similar trends to what we see in the Branin ablations.

### C.4 EFFECT OF PREFIX SEQUENCES

During the evaluation, the unrolled trajectory depends on two factors affecting the unrolled output: Evaluation Regret Budget $\hat{R}$ and the prefix sequence. Empirically, we found that $\hat{R}$ has a larger impact on the unrolled trajectory than the prefix sequence. To show this, we first evaluate the Branin task for 10 different randomly sampled prefix sequences for a fixed $\hat{R}$ and then do the same with

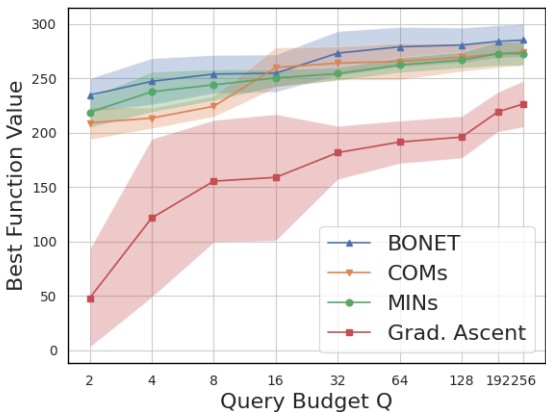

Figure 12: Results for various query budget values $Q$ for D'Kitty task, averaged over 5 runs. We match or outperform other baselines on almost all the values of $Q$.

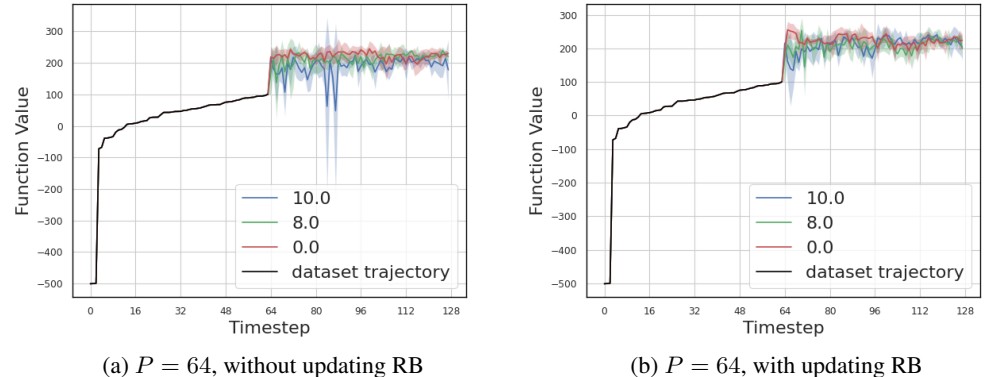

(a) $P = 64$, without updating RB

(b) $P = 64$, with updating RB

Figure 13: Figures 13a and 13b show plots of the trajectories generated on DKitty for different values of evaluation RB (0, 8 and 10). In Figure 13a we show results without updating RB, and in Figure 13b we show results with updating. All the trajectories are averaged over 5 runs.

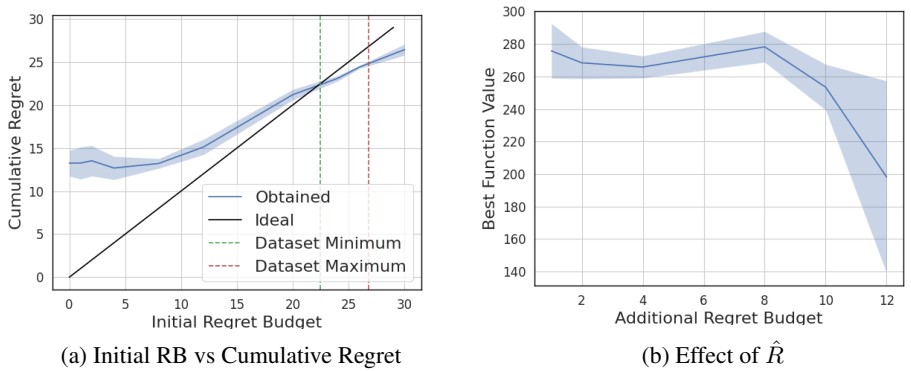

(a) Initial RB vs Cumulative Regret

(b) Effect of $\hat{R}$

Figure 14: Ablations on D'Kitty, averaged over 5 runs.

10 different $\hat{R}$ values sampled from the range $(0.0, 0.5)$ for the same prefix sequence. Figure 15 shows the standard deviation of the minimum regret of the 10 different unrolled trajectories for 3

Table 7: Unnormalized **100th percentile** results

| BASELINE | TFBIND8 | TFBIND10 | SUPERCONDUCTOR | ANT | D'KITTY | CHEMBL | NAS |
|---|---|---|---|---|---|---|---|
| $\mathcal{D}$ (best) | 0.439 | 0.00532 | 74.0 | 165.326 | 199.231 | 443000.000 | 63.79 |
| CbAS | $0.958 \pm 0.018$ | $0.761 \pm 0.067$ | $83.178 \pm 15.372$ | $468.711 \pm 14.593$ | $213.917 \pm 19.863$ | $389000.000 \pm 500.000$ | $66.355 \pm 0.79$ |
| GP-qEI | $0.824 \pm 0.086$ | $0.675 \pm 0.043$ | $\mathbf{92.686 \pm 3.944}$ | $480.049 \pm 0.000$ | $213.816 \pm 0.000$ | $387950.000 \pm 0.000$ | $\mathbf{69.722 \pm 0.59}$ |
| CMA-ES | $0.933 \pm 0.035$ | $\mathbf{0.848 \pm 0.136}$ | $90.821 \pm 0.661$ | $\mathbf{1016.409 \pm 906.407}$ | $4.700 \pm 2.230$ | $388400.000 \pm 400.000$ | $\mathbf{69.475 \pm 0.79}$ |
| Gradient Ascent | $\mathbf{0.981 \pm 0.010}$ | $0.770 \pm 0.154$ | $\mathbf{93.252 \pm 0.886}$ | $-54.955 \pm 33.482$ | $226.491 \pm 21.120$ | $390050.000 \pm 2000.000$ | $63.772 \pm 0.000$ |
| REINFORCE | $0.959 \pm 0.013$ | $0.692 \pm 0.113$ | $89.027 \pm 3.093$ | $-131.907 \pm 41.003$ | $-301.866 \pm 246.284$ | $388400.000 \pm 2100.000$ | $39.724 \pm 0.000$ |
| MINs | $0.938 \pm 0.047$ | $0.770 \pm 0.177$ | $89.469 \pm 3.227$ | $533.636 \pm 17.938$ | $272.675 \pm 11.069$ | $390950.000 \pm 200.000$ | $66.076 \pm 0.46$ |
| COMs | $0.964 \pm 0.020$ | $0.750 \pm 0.078$ | $78.178 \pm 6.179$ | $540.603 \pm 20.205$ | $\mathbf{277.888 \pm 7.799}$ | $390200.000 \pm 500.000$ | $64.041 \pm 1.390$ |
| BONET | $\mathbf{0.975 \pm 0.004}$ | $\mathbf{0.855 \pm 0.139}$ | $80.84 \pm 4.087$ | $\mathbf{567.042 \pm 11.653}$ | $\mathbf{285.110 \pm 15.130}$ | $\mathbf{391000.000 \pm 1900.000}$ | $66.779 \pm 0.16$ |

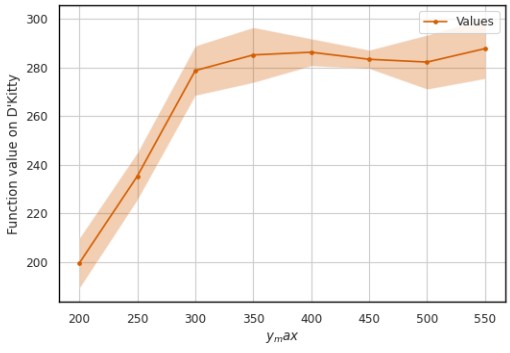

Figure 15: Standard deviation of the minimum regret of the unrolled trajectories with 1) 10 different prefix sequences for a fixed $\hat{R}$ and 2) 10 different $\hat{R}$ for a fixed prefix sequence.

fixed $\hat{R}$ and prefix sequences, respectively. The standard deviation for the variation in prefix length is consistently lower than that for the variation in $\hat{R}$, explaining our choice of spending query budget on different $\hat{R}$ rather than different prefix sequences.

## C.5 EFFECT OF ESTIMATING $y_{\mathrm{MAX}}$

A key assumption of our method is the knowledge of $y_{\mathrm{max}}$. Though in many problems this is not an issue, there are many other problems where the value of $y_{\mathrm{max}}$ may not be known. A simple solution could be to just estimate $y_{\mathrm{max}}$. In Figure 16 we evaluate BONET on D'Kitty multiple varying values of $y_{\mathrm{max}}$ starting from just beyond the dataset maxima. We find that the value of $y_{max}$ initially affects performance alot, but beyond a point, it plateaus..

Figure 16: Best points for different values of $y_{\mathbf{max}}$ on D'Kitty.

Table 8: Results for when a random $x\%$ subsection of the offline dataset was withheld during training from BONET

|  | 10% | 50% | 90% | 99% |
|---|---|---|---|---|
| $\mathcal{D}$ (best) | 74.21 | 74.20 | 73.99 | 65.71 |
| BONET | $286.60 \pm 1.47$ | $284.74 \pm 23.68$ | $274.11 \pm 7.57$ | $241.17 \pm 18.07$ |

Table 9: Results for when the top $x\%$ of the offline dataset was withheld during training from BONET

|  | 10% | 50% | 90% | 99% |
|---|---|---|---|---|
| $\mathcal{D}$ (best) | 61.14 | $-40.40$ | $-545.36$ | $-548.89$ |
| BONET | $267.68 \pm 2.28$ | $261.04 \pm 28.09$ | $211.56 \pm 16.74$ | $193.27 \pm 5.51$ |

### C.6 ABLATION ON MODEL PARAMETERS

In this experiment we study the effect of changing the number parameters in BONET. We do this by altering the number of layers and heads in BONET on D'Kitty. We find that increasing the number of parameters helps up to a point, beyond which the model over-fits. It is important to note that we present this study only to understand the impact of model size on our performance. We don't actually tune over these parameters in our experiments. They are kept fixed across all the discrete and continuous tasks (refer to Table 5).

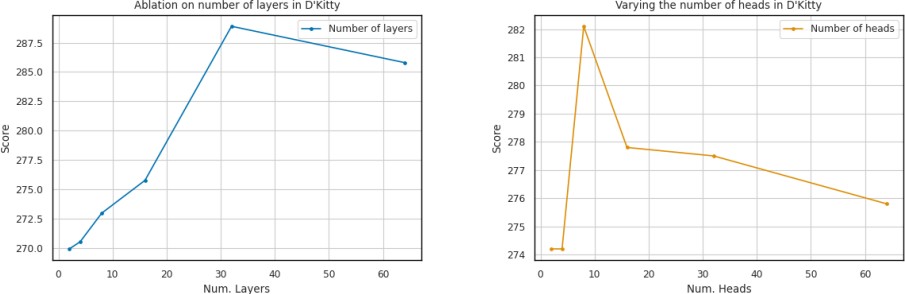

(a) Varying the number of layers in BONET. Heads are fixed at 8

(b) Varying the number of heads in BONET. Layers are fixed at 16

Figure 17: We show the performance of various models with differing number of layers and heads on D'Kitty to see their effect on BONET. We find that increasing the number of parameters helps upto a point, beyond which it overfits.

### C.7 ABLATIONS ON DATASET SIZE

To test the limits of BONET we run experiments where we withhold offline training data from BONET and evaluate the performance. We have two settings, one where we withhold an $x\%$ size random subsection of data in Table 8, and another where we withhold the top $x$ percentile of data during training and evaluation in Table 9. We see that with just reducing the number of points, we don't see as sharp of a drop off in performance as compared to when we withhold the good points in the dataset. This leads us to believe that the dominating effect is not the size of the dataset exactly, but the quality of points in the dataset. Further, note that even here the points proposed are significantly larger than the maximum point in the dataset, which rules out the possibility of memorization for BONET.

## C.8 Noise Ablation

We run an experiment where we add progressive larger amounts of noise to the $y$ values in our offline dataset while training our model, to test how robust BONET is to noisy data. We report the results in Table 10 for D'Kitty. We find that, as expected, increasing noise reduces performance, and BONET is in fact reasonably robust to noise, and the largest drop-off occurs when the magnitude of noise is equal to the magnitude of values.

Table 10: Ablation on adding various magnitudes of noise to training data

| NOISE SCALE | SCORE |
|---|---|
| 0% of max | 285.110 |
| 2% of max | 279.746 |
| 20% of max | 255.925 |
| 100% of max | 137.485 |

## C.9 Random Baseline

One might argue that BONET just memorizes the best points in the offline dataset and outputs random points close to those best points during evaluation. To rule out this possibility, we perform a simple experiment for the D'Kitty task. We choose a small hypercube domain around the optimal point in the offline dataset and uniformly randomly sample 256 points in that domain. In Table 11, we show the results for different widths of this hypercube. 0 width means only the best point in the dataset.

For smaller hypercubes around the best points in the offline dataset, we see that the best point found by 256 random searches is roughly 225, which is significantly lower than what BONET finds (291.08). For larger hypercubes, the points are highly diverse. These observations suggest that this optimization problem cannot be solved by just randomly outputting points around the best point in the dataset. If we look at the 256 points output by BONET, they are consistently good (mean is 220), with comparatively very low variance. This suggests that BONET is not simply outputting random points around the best points in the dataset.

# D Additional Analysis

## D.1 Vizualization of Predicted Points

Here we try to visualize the predicted points of BONET compared to the points in the offline data to study the nature of the points proposed by the model. As shown in Figure 18, BONET generalizes well on the unseen maxima regions of the function and produces low regret points.

## D.2 Active GP experiment

We also run a experiment to compare BONET with an active BBO method. Namely, we compare BONET with active BayesOpt, using the same GP prior and acquisition function (quasi-Expected Improvement) as mentioned in Section 3. The difference between the active method and the offline method we compare with in Table 2 is that while the active method directly optimizes the ground truth function, the offline method first trains a surrogate neural network on the data, and then performs bayesian optimization on the surrogate instead of the ground truth function. This is done to make the BayesOpt baseline fully offline, and is the same procedure followed by Trabucco et al. (2022; 2021). Note that this would result in an unfair comparison since the method is both online and queries the oracle function resulting in it using more queries than our budget. As shown in Table 12, We find that using oracle actually doesn't necessarily improve performance across all tasks, and there are other tasks where the performance doesn't change at all. And our model does outperform even the oracle GP-qEI method on several tasks.

Table 11: Results of using a simple sampling strategy randomly from a small hypercube centered around the optima. We find that BONET considerably beats this baseline, indicating that generalization occurring with BONET is not fortuitous.

| Width of hypercube | Max. function Value | Mean function Value | Std. Deviation |
|---|---|---|---|
| 0.0 | 199.23 | 199.23 | 0.0 |
| 0.005 | 212.66 | 190.68 | 9.28 |
| 0.01 | 222.44 | 182.13 | 12.21 |
| 0.05 | 226.10 | $-169.62$ | 331.00 |
| 0.1 | 209.00 | $-368.71$ | 261.37 |
| BONET | 291.08 | 221.00 | 24.43 |

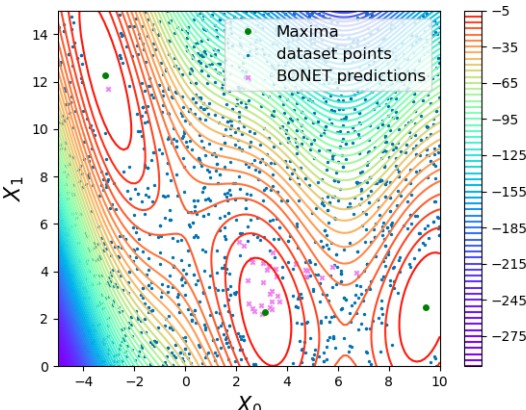

Figure 18: Vizualization of 32 points unrolled by a sample evaluation trajectory of BONET compared to 2000 points randomly sampled from the offline dataset. Three maxima regions don't contain any dataset points because of the removal of the top 10%-ile of uniformly sampled points, as described in the section 3.1. However, almost all the unrolled points fall into a maxima region, clearly showing the generalization capability of BONET.

Table 12: Comparison with GP-qEI on oracle function

| | TFBIND8 | TFBIND10 | SUPERCONDUCTOR | ANT | D'KITTY |
|---|---|---|---|---|---|
| GP-qEI (active) | $0.945 \pm 0.018$ | $0.922 \pm 0.231$ | $94.587 \pm 2.137$ | $480.049 \pm 0.000$ | $213.816 \pm 0.000$ |
| GP-qEI | $0.824 \pm 0.086$ | $0.675 \pm 0.043$ | $92.686 \pm 3.944$ | $480.049 \pm 0.000$ | $213.816 \pm 0.000$ |
| BONET | $0.975 \pm 0.004$ | $0.855 \pm 0.139$ | $80.84 \pm 4.087$ | $567.042 \pm 11.653$ | $285.110 \pm 15.130$ |

### D.3 T-SNE PLOTS ON D'KITTY

We show t-SNE plots on DKitty for the datasets of differing sizes, with the removal of randomly sampled $x\%$ of the data (setting one described in the previous section). The blue points represent points proposed by our model, and the red points represent points in the dataset. In general, blue points do not overlap the red points, indicating that the points proposed by BONET are from a different region.

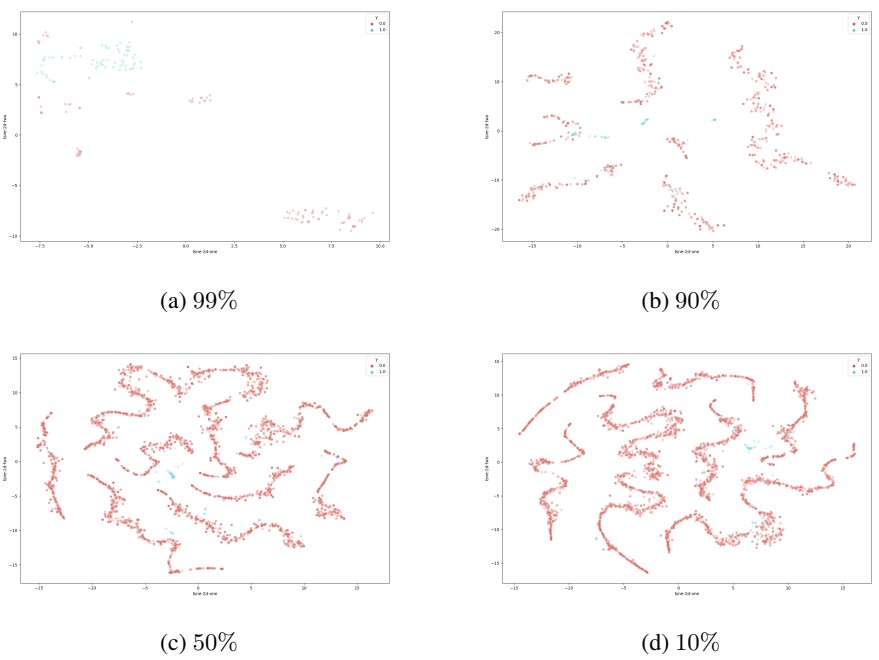

(a) 99%

(b) 90%

(c) 50%

(d) 10%

Figure 19: We show tSNE plots on DKitty for the datasets of differing sizes with the removal of randomly sampled $x\%$ of the data. The blue points represent points proposed by our model, and the red points represent points in the dataset. We find that in general, blue points do not overlap the red points, indicating that the points proposed by BONET are from a different region.

