# OpenReview forum: "Generative Pretraining for Black-Box Optimization"
_ICLR.cc/2023/Conference — Submitted to ICLR 2023_

### Official Review · Reviewer_oyMp · 2022-10-21

**Confidence:** 4
**Correctness:** 2
**Technical Novelty And Significance:** 2
**Empirical Novelty And Significance:** 2
**Recommendation:** 5

**Clarity, Quality, Novelty And Reproducibility:**

The paper's writing is generally clear and easy to understand - please see my comments above.

Regarding the paper's quality, I have various concerns regarding the proposed method and the experimental evaluation - please see my comments above.

I think the main idea of the paper, learning the trajectory towards the optimal value using offline dataset, is instersting and new - please see my comments above.

The code is made available so I believe in the reproducibilty of the paper.

**Strength And Weaknesses:**

Strengths:
+ Overall, I think the idea of learning the trajectory towards the optimal value using offline dataset is interesting and might have potential.
+ The paper's writing is also clear and easy to understand.
+ The experiments include real-world problems.

Weaknesses:
I have various concerns regarding the paper. Please see my comments below.
+ The proposed sorting strategy seems to not take into account the fact that the observed data can be noisy. And the regrets in the trajectory training dataset also assume that we can know exactly the noiseless values of the observed data. But in practice, most of the time, the objective function values will be corrupted by noise. How will noise affect the performance of the proposed method?
+ The assumption of having an estimation of the objective function's optimal value f(x*) is very strong. How does the performance of the proposed method change w.r.t. this value? The sensitivity analysis conducted in Section C.5 is not enough to understand how this value affects the performance of the proposed method (it's just for one simple problem and the range of this estimated value is limited). Besides, what are the values f(x*) used in all the experiments? I seem to not be able to find this information in the main paper. Note that if the value f(x*) is known, there are other existing works developing new BO methods to use this known value to improve the efficiency of the optimization process, e.g., Knowing the what but not the where in Bayesian optimization by Nguyen and Osborne (ICML 2020).
+ A big concern is that there is a lack of comparison with warm-start BO methods, e.g., Initializing Bayesian Hyperparameter Optimization via Meta-Learning by Feurer et al (AAAI 2015). These warm-start BO methods also make use of the knowledge learned from similar datasets and then transfer this knowledge into a new optimization problem. It has been shown that these methods can help speeding up the BO process. And note that the learned datasets in these warm-start methods do not necessarily come from the same problem we need to optimize.
+ In the experiments, the offline datasets are generated from the same problem as the problem during evaluation, so I don't think the task of learning the trajectory towards the optimal value is challenging anymore. For the Branin function, the offline datasets are constructed from 5000 data points in the domain, this is really big, and to me, this offline dataset can already help to learn the objective function quite correctly. For other real-world tasks of Design-Bench, I'm not sure if I missed it but I can't find information regarding the size of the offline datasets of these tasks in the main paper. How much data used in the offline datasets of these tasks?
+ In Tables 2 & 3, what do 100th percentile comparative evaluation and 50th percentile evaluations mean?


**Summary Of The Paper:**

The paper proposes a new generative framework to pretrain the trajectory towards the optimal value of an objective function on an offline dataset, then use this generative model (more specifically, an autoregressive model) in evaluation to select data points so as to find the optimal value of the objective function. To synthesize the trajectories from the offline data, the paper presents a sampling strategy based on a simple heuristic of sorting the function values of the data points in the offline dataset and performing a binning process that weighs more on data points with higher function values. Experiments are conducted on a synthetic function and various real-world problems to understand the performance of the proposed approach.

**Summary Of The Review:**

Even though I think the main idea of the paper is interesting and new, I have various serious concerns regarding the rigorousness and soundness of the proposed method, and also the experimental evaluation. Please see my comments above.

---

> ### Author Response · Authors · 2022-11-16
> **Author response**
>
> We thank the reviewer for their feedback and analysis of our work. We appreciate that the reviewer found our work to be well written and novel. We have tried to address the concerns raised by the reviewer below and incorporate the suggestions made.
>
> Q. In Tables 2 & 3, what do 100th percentile comparative evaluation and 50th percentile evaluations mean?
>
> A. We follow the convention used in Design-Bench, and related works to compare the $Q$ predicted points (the budget is $Q$) by the models. 100th percentile comparison is essentially comparing the maxima (100th percentile) of the Q predictions. Similarly for the 50th percentile results, we are comparing the median (50th percentile) of the Q predictions made by the models.
>
> Q. In the experiments, the offline datasets are generated from the same problem as the problem during evaluation, so I don't think the task of learning the trajectory towards the optimal value is challenging anymore. For the Branin function, the offline datasets are constructed from 5000 data points in the domain, this is really big, and to me, this offline dataset can already help to learn the objective function quite correctly.
>
> A. Our experimental protocol during training and testing is consistent with prior works in offline optimization. We also emphasize that the Branin experiment is an illustrative example on a two dimensional domain for visualization and probing of the different components of BONET. That said, it is still challenging to learn as we made careful efforts in removing the top 10% of points from the dataset. This was mentioned in Section 3.1 and can also be seen in the large difference between the optima and the dataset maximum in Table 1. For Design-Bench, we have added a table to Appendix B.6 with details on the size of the dataset for each task. The dataset sizes range from 1000-30000 data points, and the dimensionalities range from 8-86.
>
> Q. A concern is that there is a lack of comparison with warm-start BO methods.
>
> A. We would like to emphasize that we are in the fully offline setting, whereas the warm start methods cited above work in a hybrid setting which allows online querying in addition to having offline data. Hence, these works are not comparable. We also have an experiment in Appendix D.2 where we run a BayesOpt experiment which fits a GP to a large portion of the offline data and runs active BO with qEI acquisition function.
>
> Q. The assumption of having an estimation of the objective function's optimal value $f(x^*)$ is very strong. How does the performance of the proposed method change w.r.t. this value? The sensitivity analysis conducted in Section C.5 is not enough to understand how this value affects the performance of the proposed method (it's just for one simple problem and the range of this estimated value is limited).
>
> A. Thank you for pointing out this paper. We have cited it in the updated draft. However, this is not directly applicable to our setting, since they work in the online setting. We agree that the assumption that we know $f(x^*)$ may not hold everywhere, but as highlighted in [1], it does hold for a large class of problems in practice. Moreover, we find that our approach is relatively robust to conditioning on different values larger than the maximum value, as we see in our expanded analysis in Appendix C.5. We have expanded our analysis in Appendix C.5 to include more values of $y_{max}$, ranging from close to the dataset optima (200) onwards. We find that as we increase the assumed value of $y_{max}$, we do see an improvement in performance, but beyond a point the performance plateaus. This is different to what [1] notice, they find that any values that stray from the optima (below or above) cause performance issues, whereas with our method, only values lower than the optima cause issues, higher values still perform similarly to the optima.
> We have added a table in Appendix B.6 with the choices of f(x*). Appendix B.6 also has additional information about the Design-Bench tasks.
> [1] Knowing the what but not the where in Bayesian optimization by Nguyen and Osborne (ICML 2020).
>
> Q. The proposed sorting strategy seems to not take into account the fact that the observed data can be noisy. And the regrets in the trajectory training dataset also assume that we can know exactly the noiseless values of the observed data. But in practice, most of the time, the objective function values will be corrupted by noise. How will noise affect the performance of the proposed method?
>
> A. We run an experiment where we add noise to the function evaluations of D’Kitty. We test with two noise levels. Here are the results. We find that with only a little noise, the performance isn’t affected too much. Even a significant fraction like 20% still sees only a moderate decline in performance.
>
> | Noise level | Score |
> | ----------- | ----------- |
> | 0% of max | 285.110 |
> | 2% of max | 279.746 |
> | 20% of max | 255.925 |
> | 100% of max | 137.485 |

---

> > ### Author Response · Authors · 2022-11-17
> > **Response**
> >
> > Thank you again for your constructive feedback on our paper. We believe our reply addresses the questions raised in the review. As we near the end of the discussion period, please let us know if there are any further concerns and we are happy to address them in the remaining time.

---

> > > ### Comment · Reviewer_oyMp · 2022-12-02
> > > **Response after rebuttal**
> > >
> > > Dear authors,
> > >
> > > Thank you for your rebuttal and sorry for the late response my from my side. I have read the authors' response but I still have various concerns regarding the proposed method. First, the assumption of having an estimation of the objective function's optimal value f(x*) is very strong, I saw the authors have added more experimental results but only with one dataset, however, I expect this to be investigated more thorough in order to illustrate the merit of the proposed method. Secondly, I still have concerns regarding the use of large amount of dataset offline and the comparison with warm-start related approaches to show that the proposed method still outperforms the case when we make use of this offline dataset in the BO process.
> > >
> > > For these reasons, I decided to still keep my score at 5.

---

> > > > ### Author Response · Authors · 2022-12-05
> > > > **Response**
> > > >
> > > > We thank the reviewer for their response. We try and address any further questions here.
> > > >
> > > > **Investigate knowledge of** $f(x^*)$
> > > >
> > > > Due to time and compute constraints, we showed the ablation results on 1 dataset originally. We have now run the same experiment on AntMorphology, the results are in the following table. To summarize, we see a similar trend to what we see in DKittyMorphology, where small $y_{max}$ values can hamper performance slightly, but larger ones do not affect the score as much.
> > > >
> > > > | $y_{max}$      | Score |
> > > > | ----------- | ----------- |
> > > > | 400      | $543.629$       |
> > > > | 500   | $546.212$        |
> > > > | 600   | $561.016$        |
> > > > | 700   | $572.996$        |
> > > >
> > > > **Comparison with warm-start approaches**
> > > >
> > > > As mentioned earlier, this is an unfair comparison as the warm-start approaches are designed for finetuning in the online setting. In our case, we work in the purely offline setting.
> > > >
> > > > **Size of offline dataset**
> > > >
> > > > The size of datasets for Design-Bench is standard and not set by us. All the baselines use the same size of datasets. For Branin, we ran an additional experiment with only 50 randomly sampled points, and we achieved a score of $-2.13 \pm 0.15$, slightly worse than with 5000 points. This performance is still much better than the dataset maximum.
> > > >
> > > > | Size      | $\mathcal{D}_{max}$ | BONET | Grad. Ascent |
> > > > | ----------- | ----------- | ----------- | ----------- |
> > > > | 50      | $-6.231$       | $-2.13 \pm 0.15$ | $-4.64 \pm 3.17$ |
> > > > | 5000   | $-6.199$        | $-1.79 \pm 0.84$ | $−3.95 \pm 4.26$ |

---

### Official Review · Reviewer_GVeM · 2022-10-23

**Confidence:** 3
**Correctness:** 4
**Technical Novelty And Significance:** 2
**Empirical Novelty And Significance:** 3
**Recommendation:** 6

**Clarity, Quality, Novelty And Reproducibility:**

The paper is in general well written and easy to follow, the code is provided with an anonymous link.

**Strength And Weaknesses:**

Strength:
* The paper is well written and easy to follow.
* The proposed idea is neat and easy to use in practice.


Weakness:
* The analysis regarding \hat{R} are for the Branin task, does it hold in general?
The optimal value for \hat{R} seems to be 0 from Figure 4, how much value does it add when fitting the autoregressive model with regret information?
* In phase one, what is the choice of N_B and how it affects the performance?
* more clarification in the experimental section may improve the paper. For example, have a table of dimensionality for the 7 real world datasets, clarify the definition of normalised score in Table 2 and Table 3, and correct typos such as ''stitching'' -> ``stitching''.
* It would be interesting to see more evidence in the benefits of the model in the case of high-dimensional multi-modal scenarios. It is not clear from the experiments that the problems are high-dimensional and multi-modal.
* how does the model perform when observations are sparse?
* is there a way to extend to the case when the functional evaluations are noisy?


**Summary Of The Paper:**

The paper proposed BONET, a method to optimize expensive black-box function with offline data. BONET consists of three phases, 1) synthesize trajectories from offline data using a simple heuristic, 2) fit an autoregressive model based on the trajectories and regret budgets and 3) roll out the evaluation to output predictions for the maxima of the black-box function. Empirical study is conducted on one synthetic 2D example and 7 real-world tasks, which showed promising results.

**Summary Of The Review:**

The paper presented a neat idea in maximizing black-box functions using offline data only. Although the proposed method BONET is somehow based on exiting ideas such as importance reweighing and causally masked transformer, the simplicity of the proposed method brings value to be applicable in practice.

---

> ### Author Response · Authors · 2022-11-16
> **Author response**
>
> We thank the reviewer for their insightful comments and feedback. We appreciate that the reviewer found our work to be well written and easy to understand. We have tried to address the concerns raised by the reviewer below and incorporate the suggestions made.
>
> Q. The analysis regarding $\hat{R}$ are for the Branin task, does it hold in general? The optimal value for $\hat{R}$ seems to be 0 from Figure 4, how much value does it add when fitting the autoregressive model with regret information?
>
> A.  We present similar ablations for $\hat{R}$ for D’Kitty task in the Appendix C.3 and observe similar trends. As we see in the Figure 5 and Figures 14, regret conditioning plays an important role in generalization beyond the offline dataset maximum. While training we have not seen trajectories with RB closer to zeros, however conditioning our model with low \hat{R} indeed allows the model to generalize beyond the dataset.
>
> Q. In phase one, what is the choice of $N_B$ and how it affects the performance?
> A. We present an ablation on $N_B$ in Appendix B.1 (Figure 6b). We observe that there is a performance drop when $N_B = 1$. This is because $N_B = 1$ is equivalent to having no re-weighing.
>
> Q. more clarification in the experimental section may improve the paper. For example, have a table of dimensionality for the 7 real world datasets, clarify the definition of normalised score in Table 2 and Table 3, and correct typos such as ''stitching'' -> ``stitching''.
>
> A. We mention the dimensionality of the tasks in Section 3.2 (Lines - “D’Kitty, Ant and Superconductor are continuous tasks with dimensions 56, 60, and 86 respectively.” and “ The sequences are of length 8 (10) for TF-Bind-8 (TF-Bind-10)...”). However, we agree with the reviewer and include a Table for task related information in the Appendix B.6 (Table 6). We clarify our normalization procedure in Appendix B.6, and we use the same procedure for Tables 2 and 3. We also thank the reviewer for pointing out the typo. We have corrected it in the updated version.
>
> Q. It would be interesting to see more evidence in the benefits of the model in the case of high-dimensional multi-modal scenarios. It is not clear from the experiments that the problems are high-dimensional and multi-modal.
>
> A. Table 6 in the updated version shows that the tasks are high-dimensional. Furthermore, we refer the reviewer to the [design-bench](https://arxiv.org/pdf/2202.08450.pdf) paper which contains details on the challenges involved in these tasks.
>
> Q. how does the model perform when observations are sparse?
>
> A. We present an experiment in Appendix C.7 where we reduce the dataset size for D’Kitty using two different schemes and analyze the performance of our method. We observe that our performance doesn’t drop significantly for reduced data sizes.
>
> Q. Is there a way to extend to the case when the functional evaluations are noisy?
>
> A. We run an experiment where we add noise to the function evaluations of D’Kitty. We test with two noise levels. Here are the results. We find that with only a little noise, the performance isn’t affected too much. Even a significant fraction like 20% still sees only a moderate decline in performance.
>
> | Noise level | Score |
> | ----------- | ----------- |
> | 0% of max | 285.110 |
> | 2% of max | 279.746 |
> | 20% of max | 255.925 |
> | 100% of max | 137.485 |

---

> > ### Author Response · Authors · 2022-11-17
> > **Response**
> >
> > Thank you again for your constructive feedback on our paper. We believe our reply addresses the questions raised in the review. As we near the end of the discussion period, please let us know if there are any further concerns and we are happy to address them in the remaining time.

---

> > > ### Author Response · Authors · 2022-12-05
> > > **Followup**
> > >
> > > It's been a few weeks since we posted our rebuttal, and we sincerely hope you have had a chance to read our rebuttal and will respond to it soon. We are happy to answer any questions or concerns not yet addressed in the rebuttal.

---

### Official Review · Reviewer_Vfqb · 2022-10-24

**Confidence:** 3
**Correctness:** 3
**Technical Novelty And Significance:** 2
**Empirical Novelty And Significance:** 2
**Recommendation:** 5

**Clarity, Quality, Novelty And Reproducibility:**

This paper studies black-box optimization problems, which is not a new problem.  I have the following concerns for the authors to address:
1.	This paper assumes that offline datasets are available for pretraining, which is doubtful. It is unclear what types of black-box real-world applications satisfy this setting. The distributions of offline datasets may be different from online datasets, which deteriorates the performance of black-box optimizers.
2.	In the introduction part, the authors state that they can synthesize synthetic trajectories of offline points to mimic online points. The correctness of this statement requires further illustration. Again, the offline datasets and the online datasets may be differently distributed.
3.	Some figures are unclear to readers. For example, it is unclear what information Figures 1(b) and 1(c) convey.
4.	In the experiments, many recently proposed black-box baselines are missing. For example, the following black-box optimizers were published in top-tier conferences and should be compared with the proposed  BONET.
Explicit Gradient Learning for Black-Box Optimization
Black-Box Optimization with Local Generative Surrogates
Differentiating the Black Box: Optimization with Local Generative Surrogates

5.	On the Branin dataset, Table 1 is of little informative. Also, more baselines are required for comparison.
6.	On the DESIGN-BENCH tasks, several baselines are quite old, and Table 2 shows that the proposed BONET does not significantly outperform others. In fact, it is not the best optimizer for most tasks.
7.	Some typos should be corrected. For example,
empirical observation relating to=> empirical observation related to


**Strength And Weaknesses:**

Strength:
1.	The idea to pre-train a generative model is interesting.
2.	The paper is well-structured.
Weakness:
1.	The setting that offline datasets are available is doubtful.
2.	Experiments are not convincing enough to illustrate the effectiveness of the proposed framework.
3.	Some statements are unclear and difficult to understand.



**Summary Of The Paper:**

This paper studies black-box optimization and proposes BONET, which is a generative pre-trained model from offline datasets. An autoregressive model on fixed-length trajectories is trained, and a sampling strategy is designed to synthesize trajectories from offline data using a simple heuristic of rolling out monotonic transitions.

**Summary Of The Review:**

This paper proposes an interesting method to deal with black-box optimization problems. However, it is unclear what real-world applications are suitable where off-line datasets are suitable for black-box optimization, the proposed method did not outperform baselines in the experiments, and some state-of-the-arts were not compared.

---

> ### Author Response · Authors · 2022-11-15
> **Response**
>
> We thank the reviewer for their insightful comments and feedback. We appreciate that the reviewer found our work interesting and well-structured.
>
> Q1. This paper assumes that offline datasets are available for pretraining, which is doubtful. It is unclear what types of black-box real-world applications satisfy this setting. The distributions of offline datasets may be different from online datasets, which deteriorates the performance of black-box optimizers.
>
> A1. We respectfully disagree and would argue that the offline setting for black-box optimization is well studied in the literature. [mins], [coms], are some of the relevant works which develop methods for the same setting. [design-bench] motivates each of its tasks with real-world applications that span materials, protein, and molecule engineering. In particular, this setting is useful when the underlying black-box function is very expensive to evaluate and we have past evaluations for it.
>
> Q2. In the introduction part, the authors state that they can synthesize synthetic trajectories of offline points to mimic online points. The correctness of this statement requires further illustration. Again, the offline datasets and the online datasets may be differently distributed.
>
> A2. We would like to emphasize here that we did not claim to exactly reproduce the trajectories of online optimizers. This is difficult because of the reason mentioned by the reviewer – the distribution of online and offline datasets may be different. Instead, we only try to mimic a specific behavior of the online trajectories. In particular, the moving average of the function values of the queried points is nearly monotonic with time. We use a simple sorting heuristic to approximate this behavior.
>
> Q3. Some figures are unclear to readers. For example, it is unclear what information Figures 1(b) and 1(c) convey.
>
> A3. In Figure 1(b), we illustrated sample synthetic trajectories that are constructed using our SORT-SAMPLE algorithm (dotted lines) and the trajectories unrolled by the model (solid line). The unrolled model trajectory starts with points having low function values (blue points) and eventually go to the maxima regions (red points). In Figure 1(c) we show trajectories output by a simple online optimizer (BayesOpt using Gaussian Process) on three different standard tasks. We show this plot to provide evidence that the online trajectories have increasing function values. We will expand on these explanations in the paper.
>
> Q4. In the experiments, many recently proposed black-box baselines are missing. For example, the following black-box optimizers were published in top-tier conferences and should be compared with the proposed BONET. Explicit Gradient Learning for Black-Box Optimization Black-Box Optimization with Local Generative Surrogates Differentiating the Black Box: Optimization with Local Generative Surrogates
>
> A4. We would like to remind the reviewer that we are working in the offline setting where we do not have active access to the black-box function during training. All the works mentioned by the reviewer are in the online setting where we can actively query the function. Thus, these baselines are not relevant to our work.
>
> Q5. On the Branin dataset, Table 1 is of little informative. Also, more baselines are required for comparison.
>
> A5. As mentioned in the paper, Branin is an illustrative toy task to visualize our method and probe its various components. It is a 2-dimensional task where benchmarking is of little importance but rather the systematic exploration and visualization of design choices provide empirical validation of our approach. We also included Gradient Ascent as a representative forward approach to show how such naive approaches can be suboptimal even in these 2d tasks. For benchmarking on high-dimensional real-world  domains, we have included the DESIGN-BENCH suite of tasks.
>
> Q6. On the DESIGN-BENCH tasks, several baselines are quite old, and Table 2 shows that the proposed BONET does not significantly outperform others. In fact, it is not the best optimizer for most tasks.
>
> A6. We would like to note here that the Design-Bench paper was released in 2021 which contained the similar set of baselines that we present. BONET is amongst the top 2 in 5 out of the 7 tasks. Furthermore, it has the best average rank among all the methods. There is no other baseline that is a clear winner even if we remove BONET from the table.
>
> Q7. Some typos should be corrected. For example, empirical observation relating to=> empirical observation related to
>
> A7. Thanks for pointing it out. We have replaced ‘relating’ with ‘related’ in the updated version.

---

> > ### Author Response · Authors · 2022-11-17
> > **Response**
> >
> > Thank you again for your constructive feedback on our paper. We believe our reply addresses the questions raised in the review. As we near the end of the discussion period, please let us know if there are any further concerns and we are happy to address them in the remaining time.

---

> > > ### Author Response · Authors · 2022-12-05
> > > **Followup**
> > >
> > > It's been a few weeks since we posted our rebuttal, and we sincerely hope you have had a chance to read our rebuttal and will respond to it soon. We are happy to answer any questions or concerns not yet addressed in the rebuttal.

---

### Official Review · Reviewer_nSto · 2022-11-01

**Confidence:** 2
**Correctness:** 4
**Technical Novelty And Significance:** 2
**Empirical Novelty And Significance:** 2
**Recommendation:** 5

**Clarity, Quality, Novelty And Reproducibility:**

"We haven’t included Hopper since the domain is buggy - we found that the oracle function used to evaluate
the task was highly inaccurate and noisy." Would the author please comment on why noise cannot be modeled and accounted for? For example, in a Gaussian process framework, observation noise can be incorporated. This is especially important since many real-world problems are corrupted with noise.

**Strength And Weaknesses:**

Instead of a  heuristic for transitioning between low and high fidelity, it is interesting to see if the transition can be modeled and learned.

It is also interesting to see how this approach compares to multi-task learning where different tasks include different data distributions amongst offline and online datasets.
 This approach seems to have commonalities with Thompson Sampling (TS) and batch Thompson Sampling (q-TS) from Bayesian optimization literature. Although this work cites the related papers, a more through comparison/analysis can be valuable.


**Summary Of The Paper:**

This paper focuses on the optimization of a black-box expensive to evaluate function in the low data regime where the goal is to take advantage of offline related datasets. Specifically, when the offline and online data have different distributions, this work proposes to build a a generative model for pretraining a novel black-box optimizer based on the offline data.

**Summary Of The Review:**

This paper addresses and interesting problem. However, comparison to other related works can be improved. Without such comparison, an accurate judgement is hard to make.

---

> ### Author Response · Authors · 2022-11-16
> **Author response**
>
> We thank the reviewer for their insightful comments and feedback. We appreciate that the reviewer found our work interesting and well-structured. We have tried to address the concerns raised by the reviewer below and incorporate the suggestions made.
>
> Q. "We haven’t included Hopper since the domain is buggy - we found that the oracle function used to evaluate the task was highly inaccurate and noisy." Would the author please comment on why noise cannot be modeled and accounted for? For example, in a Gaussian process framework, observation noise can be incorporated. This is especially important since many real-world problems are corrupted with noise.
>
> A. We have a more detailed discussion on why we exclude Hopper in Appendix B.7. In short, we find that the signal-to-noise ratio of the function value in Hopper to be very low, meaning in many cases the noise overpowers the signal. For instance, one of the points in the dataset has a function value of ~1400, but we find that on evaluating the point with the oracle 20 times, and taking the mean value, it only has a value of ~600. Furthermore, the creators of Design-Bench themselves acknowledge this point in this pull request and are working on fixing this issue.

---

> > ### Author Response · Authors · 2022-11-17
> > **Response**
> >
> > Thank you again for your constructive feedback on our paper. We believe our reply addresses the questions raised in the review. As we near the end of the discussion period, please let us know if there are any further concerns and we are happy to address them in the remaining time

---

> > > ### Author Response · Authors · 2022-12-05
> > > **Followup**
> > >
> > > It's been a few weeks since we posted our rebuttal, and we sincerely hope you have had a chance to read our rebuttal and will respond to it soon. We are happy to answer any questions or concerns not yet addressed in the rebuttal.

---

### Author Response · Authors · 2022-11-18
**Summary of changes**

We thank all the reviewers for their insightful questions and feedback. We are pleased that many of the reviewers found our paper clear, and easy to follow.  We are also happy to acknowledge that reviewer GVeM found our work to bring value to practical scenarios due to its simplicity. We are pleased that reviewers nSto, VFqb and oyMp found our contributions to be interesting. Following suggestions from the reviewers, we have made a few changes to our paper:
* We have expanded on the experiment in Appendix C.5 with more data points
* We have cleaned up some of the writing and typos.
* We have added an additional ablation on noise to Appendix C.8

We believe these changes address all of the reviewers’ primary concerns and we have further replied to each reviewer on specific clarifications.

---

### Author Response · Authors · 2022-11-30
**Rebuttal reminder**

Dear reviewers and AC,

It's been a few weeks since we posted our rebuttal, and we sincerely hope that the reviewers have had a chance to read our rebuttal and will respond to it soon. We are happy to answer any questions or concerns not yet addressed in the rebuttal.

Thank you!

---

### Decision · Program_Chairs · 2023-01-20

**Decision:**

Reject

**Justification For Why Not Higher Score:**

This work has interesting ideas and appears well executed, but the practical relevance is unconvincing and would require some work.

**Justification For Why Not Lower Score:**

N/A

**Metareview: Summary, Strengths And Weaknesses:**

Authors introduce a method to optimize expensive black-box functions with offline data. The method consists of three phases, 1) synthesize trajectories from offline data using a simple heuristic, 2) fit an auto-regressive model based on the trajectories and regret budgets and 3) roll out the evaluation to output predictions for the maxima of the black-box function. Authors report encouraging results. However, the overall concern regarding this submission is the practical relevance of the proposed approach. Detailing realistic settings would help fixing ideas and convincing the readers that generating trajectories based on offline data is sensible and useful. While results are encouraging, the paper does not include in depth ablations studies that validate the assumptions and the pertinence of the different phases (but it does contain a number of other ablations studies related to the knowledge of the approximate optimum, the inclusion of noisce, etc.).


**Summary Of Ac-Reviewer Meeting:**

N/A